# Disentangling deep neural networks with rectified linear units using duality

## Abstract

Despite their success deep neural networks (DNNs) are still largely considered as black boxes. The main issue is that the linear and non-linear operations are entangled in every layer, making it hard to interpret the hidden layer outputs. In this paper, we look at DNNs with rectified linear units (ReLUs), and focus on the gating property ('on/off' states) of the ReLUs. We extend the recently developed dual view in which the computation is broken path-wise to show that learning in the gates is more crucial, and learning the weights given the gates is characterised analytically via the so called *neural path kernel* (NPK) which depends on inputs and gates. In this paper, we present novel results to show that convolution with global pooling and skip connection provide respectively rotational invariance and ensemble structure to the NPK. To address 'black box'-ness, we propose a novel interpretable counterpart of DNNs with ReLUs namely deep linearly gated networks (DLGN): the pre-activations to the gates are generated by a deep linear network, and the gates are then applied as external masks to learn the weights in a different network. The DLGN is not an alternative architecture per se, but a disentanglement and an interpretable re-arrangement of the computations in a DNN with ReLUs. The DLGN disentangles the computations into two 'mathematically' interpretable linearities (i) the 'primal' linearity between the input and the pre-activations in the gating network and (ii) the 'dual' linearity in the path space in the weights network characterised by the NPK. We compare the performance of DNN, DGN and DLGN on CIFAR-10 and CIFAR-100 to show that, the DLGN recovers more than $83.5\%$ of the performance of state-of-the-art DNNs, i.e., while entanglement in the DNNs enable their improved performance, the 'disentangled and interpretable' computations in the DLGN recovers most part of the performance. We conclude by identifying several interesting future directions based on DLGN.

## 1 Introduction

Despite their success deep neural networks (DNNs) are still largely considered as black boxes. The main issue is that in each layer of a DNN, the linear computation, i.e., multiplication by the weight matrix and the non-linear activations are entangled. Such entanglement has its *pros* and *cons*. The commonly held view is that such entanglement is the key to success of DNNs, in that, it allows DNNs to learn sophisticated structures in a layer-by-layer manner. However, in terms of interpretability, such entanglement has an adverse effect: only the final layer is linear and amenable to a feature/weight interpretation, and the hidden layers are non-interpretable due to the non-linearities.

Prior works (Jacot et al., 2018; Arora et al., 2019; Cao & Gu, 2019) showed that training an infinite width DNN with gradient descent is equivalent to a kernel method with the so called *neural tangent kernel* matrix. As a pure kernel method, the NTK matrix performed better than other pure kernel methods. However, in relation to 'black box'-ness, there are two issues with NTK theory: (i) **Issue I:** Infinite width NTK matrix does not explain fully the success of DNNs because it was observed that finite width DNNs outperform their infinite width NTK counterparts, and it was an open question to understand this performance gap (Arora et al., 2019), and (ii) **Issue II:** Since the NTK is based on the gradients, it does not offer further insights about the inner workings of DNNs even for infinite width.

A dual view for DNNs with rectified linear units (ReLUs) was recently developed by Lakshminarayanan & Singh (2020) who exploited the gating property (i.e., 'on/off' states) of the ReLUs. The

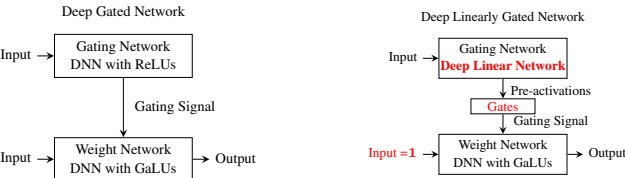

Figure 1: DGN is a setup to understand the role of gating in DNNs with ReLUs. The DLGN setup completely disentangles and re-arranges the computations in an interpretable manner. The surprising fact that a constant **1** input is given to weight network of DLGN is justified by theory and experiments in Sections 3.1 and 3.2.

dual view is essentially *linearity in the path space*, i.e., the output is the summation of path contributions. While the weights in a path are the same for each input, whether or not a path contributes to the output is entirely dictated by the gates in the path, which are 'on/off' based on the input. To understand the role of the gates, a deep gated network (DGN) (see Figure 1) was used to *disentangle* the *learning in the gates* from the *learning in weights*. In a DGN, the gates are generated (and learnt) in a 'gating network' which is a DNN with ReLUs and are applied as external signals and the weights are learnt in a 'weight network' consisting of *gated linear units* (GaLUs) (Fiat et al., 2019). Each GaLU multiplies its pre-activation and the external gating signal. Using the DGN, two important insights were provided: (i) learning in the gates is the most crucial for finite width networks to outperform infinite width NTK; this addresses **Issue I**, and (ii) in the limit of infinite width, learning the weights with fixed gates, the NTK is equal to (but for a scalar) a so called neural path kernel (NPK) which is a kernel solely based on inputs and gates. This shifts **Issue II** on interpretability to that of interpretability of the gates as opposed to interpretability of the gradients.

**Our Contribution.** We *extend the dual view* to address 'black box'-ness by completely *disentangling* the 'gating network' and the 'weight network'. Our contributions are listed below.

• **Disentangling Gating Network.** For this, we propose a novel Deep *Linearly* Gated Network (DLGN) as a *mathematically interpretable* counterpart of a DNN with ReLUs (see Figure 1). In a DLGN, the gating network is a deep linear network, i.e., there is disentanglement because of the absence of non-linear activations. The gating network is mathematically interpretable, because, the transformations from input to the pre-activations are entirely linear; we call this **primal linearity**.

• **Dual View** (Section 3.1). We present an unnoticed insight in prior work on fully connected networks that the **NPK is a product kernel** and is invariant to layer permutations. We present new results to show that (i) the **NPK is rotationally invariant** for convolutional networks with global average pooling, and (ii) the **NPK is an ensemble of many kernels** in the presence of skip connections.

• **Disentangling Weight Network.** We then argue via theory and experiments that the *weight network is disentangled in the path space*, i.e., it learns path-by-path and not layer-by-layer. For this, in Section 3.2 we show via experiments that destroying the layer-by-layer structure by permuting the layers and providing a constant **1** as input (see DLGN in Figure 1) do not degrade performance. These counter intuitive results are difficult to reconcile using the commonly held 'sophisticated structures are learnt in layers' interpretation. However, these experimental results follow from the theory in Section 3.1. In other words, it is useful to think that the learning in the weight network happens path-by-path; we call this **dual linearity**, which (for infinite width) is interpreted via the NPK.

**Message.** The DLGN is not an alternative architecture per se, but a disentanglement and an interpretable re-arrangement of the computations in a DNN with ReLUs. The DLGN disentangles the computations into two 'mathematically' interpretable linearities (i) the 'primal' linearity and (ii) the 'dual' linearity interpreted via the NPK. Using the facts that the NPK is based on input and the gates, and in a DLGN, the pre-activations in the gating network are 'primal' linear, we have complete disentanglement. We compare the performance of DNN, DGN and DLGN on CIFAR-10 and CIFAR-100 to show that, the **DLGN recovers more than** $83.5\%$ **of the performance of state-of-the-art DNNs**, i.e., while entanglement in the DNNs enable their improved performance, the 'disentangled and interpretable' computations in the DLGN recovers most part of the performance.

**Related Works.** We now compare our work with the related works.
• **Kernels.** Several works have examined theoretically as well as empirically two important kernels associated with a DNN namely its NTK based on the correlation of the gradients and the conjugate

kernel based on the correlation of the outputs (Fan & Wang, 2020; Geifman et al., 2020; Liu et al., 2020; Chen et al., 2020; Xiao et al., 2020; Jacot et al., 2018; Arora et al., 2019; Novak et al., 2018; Lee et al., 2017; 2020b). In contrast, the NPK is based on the correlation of the gates. We do not build pure-kernel method with NPK, but use it as an aid to disentangle finite width DNN with ReLUs.

● **ReLU, Gating, Dual Linearity.** A spline theory based on max-affine linearity was proposed in (Balestriero et al., 2018; Balestriero & Baraniuk, 2018) to show that a DNN with ReLUs performs hierarchical, greedy template matching. In contrast, the dual view exploits the gating property to simplify the NTK into the NPK. Gated linearity was studied in (Fiat et al., 2019) for single layered networks, along with a non-gradient algorithm to tune the gates. In contrast, we look at networks of any depth, and the gates are tuned via standard optimisers. The main novelty in our work in contrast to the above is that in DLGN the feature generation is linear. The gating in this paper refers to the gating property of the ReLU itself and has no connection to (Srivastava et al., 2015) where gating is a mechanism to regulate information flow. Also, the soft-gating used in our work and in (Lakshminarayanan & Singh, 2020) enables gradient flow via the gating network and is different from *Swish* (Ramachandran et al., 2018), which is the multiplication of pre-activation and sigmoid.

● **Finite vs Infinite Width.** Lee et al. (2020a) perform an extensive comparison of finite versus infinite width DNNs. An aspect that is absent in their work, but present in the dual view is the disentanglement of gates and weights, and the fact that the learning in gates is crucial for finite width network to outperform infinite width DNNs. In our paper, we make use of theory developed for infinite width DNNs to provide empirical insights into inner workings of finite width networks.

● **Capacity.** Our experiments on destruction of layers, and providing constant $1$ input are direct consequences of the insights from dual view theory. These are not explained by mere capacity based studies showing DNNs are powerful to fit even random labelling of datasets (Zhang et al., 2016).

## 2 PRIOR WORK : NEURAL TANGENT KERNEL AND DUAL VIEW

In this section, we will focus on the dual view (Lakshminarayanan & Singh, 2020) and how the dual view helps to address the open question in the NTK theory. We begin with a brief summary of NTK.

**NTK.** An important kernel associated with a DNN is its *neural tangent kernel* (NTK), which, for a pair of input examples $x, x' \in \mathbb{R}^{d_{in}}$, and network weights $\Theta \in \mathbb{R}^{d_{net}}$, is given by:

$$\text{NTK}(x, x') \quad = \quad \langle \nabla_\Theta \hat{y}(x), \nabla_\Theta \hat{y}(x') \rangle, \quad \text{where}$$

$\hat{y}_\Theta(\cdot) \in \mathbb{R}$ is the DNN output. Prior works (Jacot et al., 2018; Arora et al., 2019; Cao & Gu, 2019) have shown that, as the width of the DNN goes to infinity, the NTK matrix converges to a limiting deterministic matrix $\text{NTK}_\infty$, and training an infinitely wide DNN is equivalent to a kernel method with $\text{NTK}_\infty$. While, as a pure kernel $\text{NTK}_\infty$ performed better than prior kernels by more than $10\%$, Arora et al. (2019) observed that on CIFAR-10:

$$\text{CNTK-GAP: } \mathbf{77.43}\% \leq \text{CNN-GAP: } \mathbf{83.30}\%$$

where, CNN-GAP is a convolutional neural network with global average pooling and CNTK-GAP is its corresponding $\text{NTK}_\infty$ matrix. Due to this performance gap of about $5 - 6\%$, they concluded that $\text{NTK}_\infty$ does not explain fully the success of DNNs, and explaining this gap was an **open question**.

### 2.1 DUAL VIEW FOR DNNs WITH RELUs: CHARACTERISING THE ROLE OF GATES

In the dual view, the computations are broken down path-by-path. The input and the gates (in each path) are encoded in a neural path feature vector and the weights (in each path) are encoded in a neural path value vector, and the output is the inner product of these two vectors. The learning in the gates and the learning in the weights are separated in a deep gated network (DGN) setup, which leads to the two main results of dual view presented in Section 2.1.2 and Section 2.1.3, wherein, the neural path kernel, the Gram matrix of the neural path features will play a key role.

#### 2.1.1 NEURAL PATH FEATURE, VALUE, KERNEL AND DEEP GATED NETWORK

Consider a fully connected DNN with '$d$' layers and '$w$' hidden units in each layer. Let the DNN accept input $x \in \mathbb{R}^{d_{in}}$ and produce an output $\hat{y}_\Theta(x) \in \mathbb{R}$.

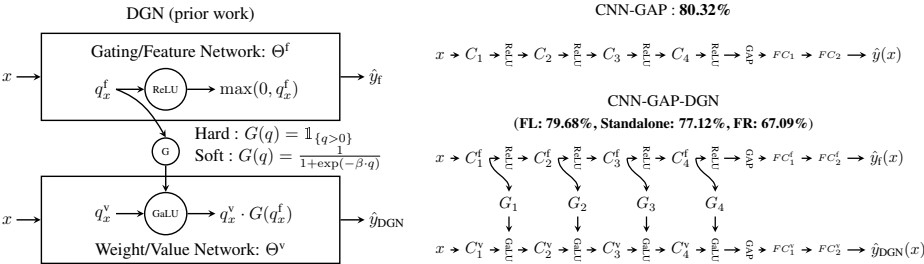

Figure 2: Shows the DGN on the left. **Training:** In the case of fixed learnt gates, the feature network is pre-trained using $\hat{y}_f$ as the output, and then the feature network is frozen, and the value network is trained with $\hat{y}_{DGN}$ as the output. In the case of fixed random gates, the feature network is initialised at random and frozen, and the value network is trained with $\hat{y}_{DGN}$ as the output. In the case of fixed gates, hard gating $G(q) = \mathbb{1}_{\{q>0\}}$ is used. **Standalone Training:** both feature and value network are initialised at random and trained together with $\hat{y}_{DGN}$ as the output. Here, soft gating $G(q) = \frac{1}{1+\exp(-\beta \cdot q)}$ is used to allow gradient flow through feature network. On the right side is the CNN-GAP and its DGN used in (Lakshminarayanan & Singh, 2020). In CNN-GAP, $C_1, C_2, C_3, C_4$ are convolutional layers, $FC_1, FC_2$ are fully connected layers. In CNN-GAP-DGN, $G_l, l = 1, 2, 3, 4$ are the gates of layers, the superscripts f, and v stand for feature and value network respectively.

**Definition 2.1.** *A path starts from an input node, passes through a weight and a hidden unit in each layer and ends at the output node. We define the following quantities for a path $p$:*

Activity : $A_\Theta(x, p)$ is the product of the '$d-1$' gates in the path.
Value : $v_\Theta(p)$ is the product of the '$d$' weights in the path.
Feature : $\phi_\Theta(x, p)$ is the product of the signal at the input node of the path and $A_\Theta(x, p)$.

*The neural path feature (NPF) given by $\phi_\Theta(x) = \left(\phi_\Theta(x, p), p = 1, \ldots, P^{fc}\right), \in \mathbb{R}^{P^{fc}}$ and the neural path value (NPV) given by $v_\Theta = \left(v_\Theta(p), p = 1, \ldots, P^{fc}\right), \in \mathbb{R}^{P^{fc}}$, where $P^{fc} = d_{in}w^{(d-1)}$ is the total number of paths.*

**Proposition 2.1.** *The output of the DNN is then the inner product of the NPF and NPV:*

$$\hat{y}_\Theta(x) = \langle \phi_\Theta(x), v_\Theta \rangle = \sum_{p \in [P]} \phi_\Theta(x, p) v_\Theta(p) \tag{1}$$

**Subnetwork Interpretation of DNNs with ReLUs.** A path is active only if all the gates in the path are active. This gives a subnetwork interpretation, i.e., for a given input $x \in \mathbb{R}^{d_{in}}$, only a subset of the gates and consequently only a subset of the paths are active, and the input to output computation can be seen to be produced by this active subnetwork. The following matrix captures the correlation of the active subnetworks for a given pair of inputs $x, x' \in \mathbb{R}^{d_{in}}$.

**Definition 2.2** (Overlap of active sub-networks). *The total number of 'active' paths for both $x$ and $x'$ that pass through input node $i$ is defined to be:*

$$\boldsymbol{overlap}_\Theta(i, x, x') \triangleq |\{p \colon p \text{ starts at node } i, A_\Theta(x, p) = A_\Theta(x', p) = 1\}|$$

**Lemma 2.1** (Neural Path Kernel (NPK)). *Let $D \in \mathbb{R}^{d_{in}}$ be a vector of non-negative entries and for $u, u' \in \mathbb{R}^{d_{in}}$, let $\langle u, u' \rangle_D = \sum_{i=1}^{d_{in}} D(i)u(i)u'(i)$. Then the neural path kernel (NPK) is given by:*

$$NPK_\Theta(x, x') \triangleq \langle \phi_\Theta(x), \phi_\Theta(x') \rangle = \langle x, x' \rangle_{\boldsymbol{overlap}_\Theta(\cdot, x, x')}$$

**Deep Gated Network (DGN)** is a setup to separate the gates from the weights. Consider a DNN with ReLUs with weights $\Theta \in \mathbb{R}^{d_{net}}$. The DGN *corresponding* to this DNN (left diagram in Figure 2) has two networks of *identical architecture* (to the DNN) namely the 'gating network' and the 'weight network' with distinct weights $\Theta^f \in \mathbb{R}^{d_{net}}$ and $\Theta^v \in \mathbb{R}^{d_{net}}$. The 'gating network' has ReLUs which turn 'on/off' based on their pre-activation signals, and the 'weight network' has gated linear units (GaLUs) (Fiat et al., 2019; Lakshminarayanan & Singh, 2020), which multiply their respective pre-activation inputs by the external gating signals provided by the 'gating network'. Since both the networks have identical architecture, the ReLUs and GaLUs in the respective networks have a one-to-one correspondence. Gating network realises $\phi_{\Theta^f}(x)$ by turning 'on/off' the corresponding GaLUs in the weight network. The weight network realises $v_{\Theta^v}$ and computes the output $\hat{y}_{DGN}(x) = \langle \phi_{\Theta^f}(x), v_{\Theta^v} \rangle$. The gating network is also called as the feature network since it realises the neural path features, and the weight network is also called as the value network since it realises the neural path value.

### 2.1.2 Learning Weights With Fixed Gates = Neural Path Kernel

During training, a DNN learns both $\phi_\Theta(x)$ as well as $v_\Theta$ simultaneously, and a finite time characterisation of this learning in finite width DNNs is desirable. However, this is a hard problem. An easier problem is to understand in a DGN, how the weights in the value network are learnt when the gates are fixed in the feature network, i.e., how $\hat{y}_{DGN}(x) = \langle \phi_{\Theta^f}(x), v_{\Theta^v} \rangle$ is learnt by learning $v_{\Theta^v}$ with fixed $\phi_{\Theta^f}(x)$. While $\hat{y}_{DGN}(x) = \langle \phi_{\Theta^f}(x), v_{\Theta^v} \rangle$ is linear in the dual variables, it is still non-linear in the value network weights $\Theta^v$. However, Lakshminarayanan & Singh (2020) showed that the dual linearity is characterised by the NPK in the infinite width regime. We state the assumption followed by Theorem 5.1 in (Lakshminarayanan & Singh, 2020), wherein, the $\text{NTK}(x, x') = \langle \nabla_{\Theta^v} \hat{y}_{\Theta_0^{DGN}}(x), \nabla_{\Theta^v} \hat{y}_{\Theta_0^{DGN}}(x') \rangle$ is due to the gradient of $\hat{y}_{DGN}$ with respect to the value network weights, and the $\text{NPK}(x, x') = \langle \phi_{\Theta_0^f}(x), \phi_{\Theta_0^f}(x') \rangle$ is due to the feature network weights.

**Assumption 2.1.** $\Theta_0^v \overset{i.i.d}{\sim} Bernoulli(\frac{1}{2})$ over $\{-\sigma, +\sigma\}$ and statistically independent of $\Theta_0^f$.

**Theorem 2.1** (Theorem 5.1 in (Lakshminarayanan & Singh, 2020)). *Under Assumption 2.1 for a fully connected DGN :*

$$NTK^{FC}(x, x') \overset{(a)}{\to} d \cdot \sigma^{2(d-1)} \cdot NPK^{FC}(x, x'), \quad as \ w \to \infty$$
$$= d \cdot \sigma^{2(d-1)} \cdot \langle x, x' \rangle \cdot \boldsymbol{overlap}(x, x')$$

**Remark.** In the fully connected case, $\boldsymbol{overlap}(i, x, x')$ is identical for all $i = 1, \ldots, d_{\text{in}}$, and hence $\langle x, x' \rangle_{\boldsymbol{overlap}(\cdot, x, x')}$ in Lemma 2.1 becomes $\langle x, x' \rangle \cdot \boldsymbol{overlap}(x, x')$ in Theorem 2.1. It follows from NTK theory that an infinite width DGN with fixed gates is equivalent to kernel method with NPK.

### 2.1.3 Learning in Gates Key For Finite Width To Be Better Than Infinite Width

The fixed gates setting is an idealised setting, in that, it does not theoretically capture the learning of the gates, i.e., the neural path features $\phi_\Theta(x)$. However, the learning in the gates can be empirically characterised by comparing **fixed learnt (FL)** gates coming from a pre-trained DNN and **fixed random (FR)** gates coming from randomly initialised DNN, and the infinite width NTK. Using a CNN-GAP and its corresponding DGN, Lakshminarayanan & Singh (2020) showed on CIFAR-10 that (see Figure 2 for details on DGN training and the CNN-GAP architecture):

FR Gates : **67**.**09**% $\leq$ CNTK-GAP: **77**.**43**% $\leq$ FL Gates: **79**.**68**% $\approx$ CNN-GAP: **80**.**32**%

based on which it was concluded that learning in the gates (i.e., neural path features) is crucial for finite width CNN-GAP to outperform the infinite width CNTK-GAP. It was also shown that the DGN can be trained **standalone** (as shown in Figure 2) and is only *marginally poor* to the DNN.

## 3 Deep Linearly Gated Networks: Complete Disentanglement

$$\text{DLGN} : x \overset{\text{Primal}}{\to} \text{Linear} \to \text{Pre-activations} \to \text{Gates} \overset{\text{lifting}}{\to} \phi_{\Theta^f}(x) \overset{\text{Dual}}{\to} \text{Linear: } \hat{y}(x) = \langle \phi_{\Theta^f}(x), v_{\Theta^v} \rangle$$

The deep linearly gated network (DLGN) has two 'mathematically' interpretable linearities, the 'primal' and the 'dual' linearities. The primal linearity is ensured in via construction and needs no theoretical justification. Once the pre-activations triggers gates, $\phi_{\Theta^f}(x)$ gets realised in the value network by activating the paths. Now, the value network itself is 'dual' linear, i.e., it simply computes/learns the inner product $\hat{y}(x) = \langle \phi_{\Theta^f}(x), v_{\Theta^v} \rangle$. Gating *lifts* the 'primal' linear computations in the feature network to 'dual' linear computations in the value network. Dual linearity is characterised by the NPK (for infinite width) which in turn depends on the input and gates, and the fact that the pre-activations to the gates are primal linear implies complete disentanglement and interpretability.

Dual linearity is mathematically evident due to the inner product relationship, however, adopting it has the following conceptual issue: it is a commonly held view that 'sophisticated features are learnt in the layers', that is, given that the input $x \in \mathbb{R}^{d_{\text{in}}}$ is presented to the value network (as in Figure 2), it could be argued that the GaLUs and linear operations are entangled which in turn enable learning of sophisticated features in the layers. In what follows, we demystify this layer-by-layer view via theory (infinite width case) in Section 3.1 , and experiments (on finite width networks) in Section 3.2, and then study the performance of DLGN in Section 3.2. The layer-by-layer view is demystified by

showing that (i) a constant **1** input can be given to the value network, (ii) layer-by-layer structure can be destroyed. The constant **1** input is meant to show that if the input is not given to the value network then it is not possible to learn sophisticated structures 'from the input' in a layer-by-layer manner. In terms of the dual linearity, providing a constant **1** input has only a minor impact, in that, the neural path feature becomes $\phi(x, p) = 1 \cdot A(x, p)$, i.e., it still encodes the path activity which is still input dependent. Since $\phi(x)$ depends only on gates, the NPK will depend only on the **overlap** matrix; results in Section 3.1 captures this in theory. Now, it could be argued that, despite a constant **1** input, the gates are still arranged layer-by-layer, due to which, the value network is still able to learn sophisticated structures in its layers. Section 3.1 has theory that points out that as long as the **correlation of the gates** is not lost, the layer-by-layer structure can be destroyed.

## 3.1 DUAL LINEARITY: NEW INSIGHTS AND NEW RESULTS

We saw in Section 2.1.2 that dual linearity is characterised by the NPK for infinite width case. In this section, we: (i) cover standard architectural choices namely convolutions with global-average-pooling and skip connections in Theorems 3.2 and 3.3; the prior result Theorem 2.1 is only for the fully connected case, (ii) present new insights on Theorem 2.1 by restating it explicitly in terms of the gates in Theorem 3.1, and (iii) discuss how the NPK structure helps in demystifying the layer-by-layer view. Note: Results in this section are about the value network and hold for both DGN and DLGN.

### 3.1.1 FULLY CONNECTED: PRODUCT OF LAYERWISE BASE KERNELS

**Theorem 3.1.** *Let $G_l(x) \in [0, 1]^w$ denote the gates in layer $l \in \{1, \ldots, d-1\}$ for input $x \in \mathbb{R}^{d_{in}}$. Under Assumption 2.1 ($\sigma = \frac{c_{scale}}{\sqrt{w}}$) as $w \to \infty$, we have for fully connected DGN/DLGN:*

$$NTK^{FC}(x, x') \to d \cdot \sigma^{2(d-1)} \cdot NPK^{FC}(x, x') = d \cdot c_{scale}^{2(d-1)} \cdot \left( \langle x, x' \rangle \cdot \Pi_{l=1}^{d-1} \frac{\langle G_l(x), G_l(x') \rangle}{w} \right),$$

• **Product Kernel : Role of Depth and Width.** Theorem 3.1 is mathematically equivalent to Theorem 2.1, which follows from the observation that **overlap**$(x, x') = \Pi_{l=1}^{(d-1)} \langle G_l(x), G_l(x') \rangle$. While this observation is very elementary in itself, it is significant at the same time; Theorem 3.1 provides the most simplest kernel expression that characterises the information in the gates. From Theorem 3.1 it is evident that the role of width is *averaging* (due to the division by $w$). Each layer therefore corresponds to a *base kernel* $\frac{\langle G_l(x), G_l(x') \rangle}{w}$ which measures the ***correlation of the gates***. The role of depth is to provide the product of kernels. To elaborate, the feature network provides the gates $G_l(x)$, and the value network realises the product kernel in Theorem 3.1 by laying out the GaLUs depth-wise, and connecting them to form a deep network. The depth-wise layout is important: for instance, if we were to concatenate the gating features as $\varphi(x) = (G_l(x), l = 1, \ldots, d-1) \in \{0, 1\}^{(d-1)w}$, it would have only resulted in the kernel $\langle \varphi(x), \varphi(x') \rangle = \sum_{l=1}^{d-1} \langle G_l(x), G_l(x') \rangle$, i.e., a *sum (not product)* of kernels.

• **Constant 1 Input.** This has a minor impact, in that, the expression on right hand side of Theorem 3.1 becomes $d \cdot c_{scale}^{2(d-1)} \cdot d_{in} \cdot \Pi_{l=1}^{d-1} \frac{\langle G_l(x), G_l(x') \rangle}{w}$, i.e., the kernel still has information of the gates.

• **Destroying structure by permuting the layers.** $\Pi_{l=1}^{d-1} \frac{\langle G_l(x), G_l(x') \rangle}{w}$ is permutation invariant, and hence permuting the layers has no effect.

### 3.1.2 CONVOLUTION GLOBAL AVERAGE POOLING: ROTATIONALLY INVARIANT KERNEL

We consider networks with circular convolution and global average pooling (architecture and notations are in the Appendix). In Theorem 3.2, let the circular rotation of vector $x \in \mathbb{R}^{d_{in}}$ by '$r$' co-ordinates be defined as $rot(x, r)(i) = x(i + r)$, if $i + r \leq d_{in}$ and $rot(x, r)(i) = x(i + r - d_{in})$ if $i + r > d_{in}$.

**Theorem 3.2.** *Under Assumption 2.1, for a suitable $\beta_{cv}$ (see Appendix for expansion of $\beta_{cv}$):*

$$NTK^{CONV}(x, x') \to \frac{\beta_{cv}}{d_{in}^2} \cdot \sum_{r=0}^{d_{in}-1} \langle x, rot(x', r) \rangle_{\textbf{\textit{overlap}}(\cdot, x, rot(x', r))}, \ as \ w \to \infty$$

• $\sum_{r=0}^{d_{in}-1} \langle x, rot(x', r) \rangle_{\textbf{overlap}(\cdot, x, rot(x', r))} = \sum_{r=0}^{d_{in}-1} \sum_{i=1}^{d_{in}} x(i) rot(x', r)(i) \textbf{overlap}(i, x, rot(x', r))$, where the inner '$\Sigma$' is the inner product between $x$ and $rot(x', r)$ weighted by **overlap** and the outer

'$\Sigma$' covers all possible rotations, which in addition to the fact that all the variables internal to the network rotate as the input rotates, results in the rotational invariance. It was observed by Arora et al. (2019) that networks with global-average-pooling are better than vanilla convolutional networks. The rotational invariance holds for convolutional architectures only in the presence of global-pooling. So, this result explains why global-average-pooling helps. That said, rotational invariance is not a new observation; it was shown by Li et al. (2019) that prediction using CNTK-GAP is equivalent to prediction using CNTK without GAP but with full translation data augmentation (same as rotational invariance) with wrap-around at the boundary (same as circular convolution). However, Theorem 3.2 is a necessary result, in that, it shows rotational invariance is recovered in the dual view as well.

- The expression in Theorem 3.2 becomes $\frac{\beta_{cv}}{d_{in}^2} \cdot \sum_{r=0}^{d_{in}-1} \sum_{i=1}^{d_{in}} \mathbf{overlap}(i, x, rot(x', r))$ for a **constant 1 input**. The key novel insight is that the rotational invariance is not lost and **overlap** matrix measures the correlation of the paths which in turn depends on the correlation of the gates.

- **Destroying structure by permuting the layers** does not destroy the rotational invariance in Theorem 3.2. This is because, due to circular convolutions all the internal variables of the network rotate as the input rotates. Permuting the layers only affects the ordering of the layers, and does not affect the fact that the gates rotate if the input rotates, and correlation in the gates is not lost.

### 3.1.3 Residual Networks With Skip Connections (ResNet): Ensemble Of Kernels

We consider a ResNet with '$(b + 2)$' blocks and '$b$' skip connections between the blocks. Each block is a fully connected (FC) network of depth '$d_{blk}$' and width '$w$'. There are $2^b$ many sub-FCNs within this ResNet (see Definition 3.1). Note that the blocks being fully connected is for expository purposes, and the result continue to hold for any kind of block.

**Definition 3.1.** *[Sub FCNs] Let $2^{[b]}$ denote the power set of $[b]$ and let $\mathcal{J} \in 2^{[b]}$ denote any subset of $[b]$. Define the '$\mathcal{J}^{th}$' sub-FCN of the ResNet to be the fully connected network obtained by (i) including $block_j, \forall j \in \mathcal{J}$ and (ii) ignoring $block_j, \forall j \notin \mathcal{J}$.*

**Theorem 3.3.** *Let $NPK_{\mathcal{J}}^{FC}$ be the NPK of the $\mathcal{J}^{th}$ sub-FCN, and $\beta_{fc}^{\mathcal{J}}$ (see Appendix for expansion of $\beta_{fc}^{\mathcal{J}}$) be the associated constant. Under Assumption 2.1, we have:*

$$NTK^{RES} \to \sum_{\mathcal{J} \in 2^{[b]}} \beta_{fc}^{\mathcal{J}} NPK_{\mathcal{J}}^{FC}, \text{ as } w \to \infty$$

- **Ensemble.** To the best of our knowledge, this is the first theoretical result to show that ResNets have an ensemble structure, where each kernel in the ensemble, i.e., $NPK_{\mathcal{J}}^{FC}$ corresponds to one of the $2^b$ sub-architectures (see Definition 3.1). The ensemble behaviour of ResNet and presence of $2^b$ architectures was observed by Veit et al. (2016), however without any concrete theoretical formalism.

- Effect of **constant 1 input** is as before for kernels $NPK_{\mathcal{J}}^{FC}$ and translates to the ensemble $NTK^{RES}$.

- **Destroying structure.** The ResNet inherits the invariances of the block level kernel. In addition, the ensemble structure allows to even remove layers. Veit et al. (2016) showed empirically that removing single layers from ResNets at test time does not noticeably affect their performance, and yet removing a layer from architecture such as VGG leads to a dramatic loss in performance. Theorem 3.3 can be seen to provide a theoretical justification for this empirical result. In other words, due to the ensemble structure a ResNet is capable of dealing with failure of components. While failure of component itself does not occur unless one makes them fail purposefully as done in (Veit et al., 2016), the insight is that even if one or many of the kernels in the ensemble are corrupt and the good ones can compensate.

## 3.2 Numerical Experiments

Section 3.1 presented theoretical results which demystified the layer-by-layer view in value network, in this section we will verify these theoretical results in experiments. We then show that DLGN recovers major part of performance of state-of-the-art DNNs on CIFAR-10 and CIFAR-100.

**Setup Details.** We consider 3 DNN architectures, C4GAP, VGG-16 and Resnet-110, and their DGN and DLGN counterparts. Here C4GAP is a simple model (achieves about $80\%$ accuracy on CIFAR-10), mainly used to verify the theoretical insights in Section 3.1. VGG-16 and Resnet-110 are

chosen for their state-of-the-art performance on CIFAR-10 and CIFAR-100. All models are trained using off-the-shelf optimisers (for more details, see Appendix D). The DGN and DLGN are trained from scratch, i.e., both the feature and value network are initialised at random and trained. In DGN and DLGN, we use soft gating (see Figure 2) so that gradient flows through the feature network and the gates are learnt (we chose $\beta = 10$). In what follows, we use the notation $\text{DGN}(x^{\text{f}}, x^{\text{v}})$ and $\text{DLGN}(x^{\text{f}}, x^{\text{v}})$ where $x^{\text{f}}$ and $x^{\text{v}}$ denote the input to the value and feature networks respectively. For instance, $\text{DLGN}(x, x)$ means that both the value and feature network of the DLGN is provided the image as input, and $\text{DLGN}(x, \mathbf{1})$ will mean that the feature network is given the image as input and the value network is given a constant $\mathbf{1}$ as input. $\text{DGN}(x, x)$ and $\text{DGN}(x, \mathbf{1})$ notation works similarly.

**Disentangling Value Network.** We show that destroying the layer-by-layer structure via permutations and providing a constant $\mathbf{1}$ input do not degrade performance. Since our aim here is not state-of-the-art performance, we use C4GAP with $4$ convolutional layers which achieves only about $80\%$ test accuracy on CIFAR-10, however, enables us to run all the $4! = 24$ layer permutations. The C4GAP, DGN and DLGN with layer permutations are shown in Figure 3. Once a permutation is chosen, it is fixed during both training and testing. The results in Table I of Figure 3 show that there is no significant difference in performance between $\text{DGN}(x, x)$ vs $\text{DGN}(x, \mathbf{1})$, and $\text{DLGN}(x, x)$ vs $\text{DLGN}(x, \mathbf{1})$, i.e., constant $\mathbf{1}$ input does not hurt. Also, there is no significant difference between the models without permutations and the models with permutations. These counter intuitive and surprising results are difficult to explain using the commonly held 'sophisticated features are learnt layer-by-layer' view. However, neither the permutations or the constant $\mathbf{1}$ input destroys the correlation in the gates, and are not expected to degrade performance as per the insights in Section 3.1. This verifies Claim I.

**DLGN Performance.** For this we choose VGG-16 and Resnet-110. The results in Table II of Figure 3 show that the DLGN recovers more than $83.5\%$ (i.e., $83.78\%$ in the worst case) of the performance of the state-of-the-art DNN. While entanglement in the DNNs enable their improved performance, the 'disentangled and interpretable' computations in the DLGN recovers most part of the performance.

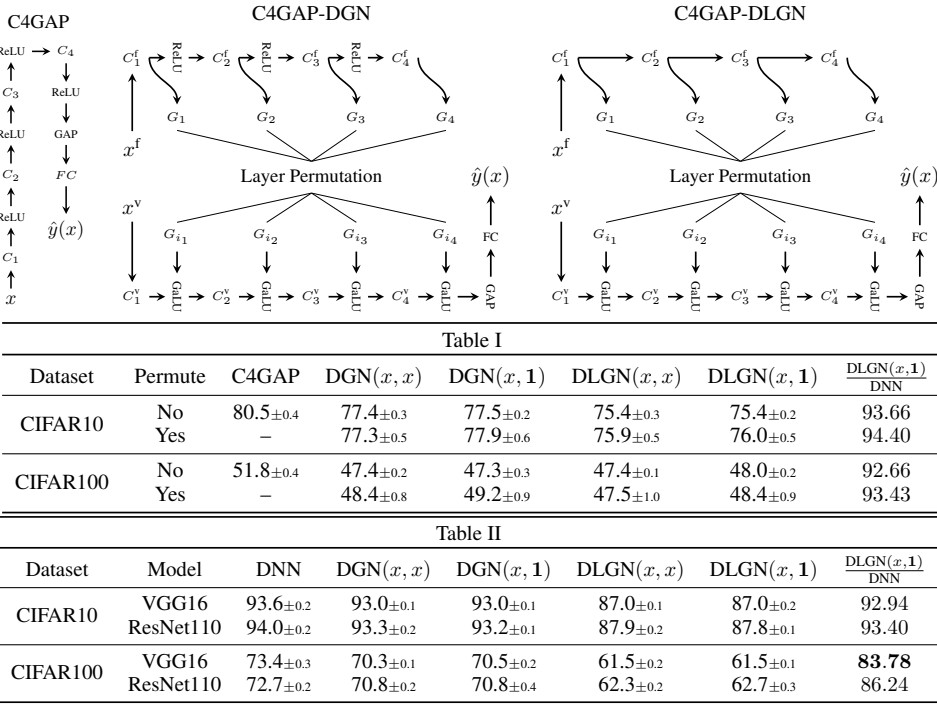

| Table I | | | | | | | |
|---|---|---|---|---|---|---|---|
| Dataset | Permute | C4GAP | $\text{DGN}(x,x)$ | $\text{DGN}(x,\mathbf{1})$ | $\text{DLGN}(x,x)$ | $\text{DLGN}(x,\mathbf{1})$ | $\frac{\text{DLGN}(x,\mathbf{1})}{\text{DNN}}$ |
| CIFAR10 | No | $80.5_{\pm 0.4}$ | $77.4_{\pm 0.3}$ | $77.5_{\pm 0.2}$ | $75.4_{\pm 0.3}$ | $75.4_{\pm 0.2}$ | 93.66 |
|  | Yes | – | $77.3_{\pm 0.5}$ | $77.9_{\pm 0.6}$ | $75.9_{\pm 0.5}$ | $76.0_{\pm 0.5}$ | 94.40 |
| CIFAR100 | No | $51.8_{\pm 0.4}$ | $47.4_{\pm 0.2}$ | $47.3_{\pm 0.3}$ | $47.4_{\pm 0.1}$ | $48.0_{\pm 0.2}$ | 92.66 |
|  | Yes | – | $48.4_{\pm 0.8}$ | $49.2_{\pm 0.9}$ | $47.5_{\pm 1.0}$ | $48.4_{\pm 0.9}$ | 93.43 |

| Table II | | | | | | | |
|---|---|---|---|---|---|---|---|
| Dataset | Model | DNN | $\text{DGN}(x,x)$ | $\text{DGN}(x,\mathbf{1})$ | $\text{DLGN}(x,x)$ | $\text{DLGN}(x,\mathbf{1})$ | $\frac{\text{DLGN}(x,\mathbf{1})}{\text{DNN}}$ |
| CIFAR10 | VGG16 | $93.6_{\pm 0.2}$ | $93.0_{\pm 0.1}$ | $93.0_{\pm 0.1}$ | $87.0_{\pm 0.1}$ | $87.0_{\pm 0.2}$ | 92.94 |
|  | ResNet110 | $94.0_{\pm 0.2}$ | $93.3_{\pm 0.2}$ | $93.2_{\pm 0.1}$ | $87.9_{\pm 0.2}$ | $87.8_{\pm 0.1}$ | 93.40 |
| CIFAR100 | VGG16 | $73.4_{\pm 0.3}$ | $70.3_{\pm 0.1}$ | $70.5_{\pm 0.2}$ | $61.5_{\pm 0.2}$ | $61.5_{\pm 0.1}$ | **83.78** |
|  | ResNet110 | $72.7_{\pm 0.2}$ | $70.8_{\pm 0.2}$ | $70.8_{\pm 0.4}$ | $62.3_{\pm 0.2}$ | $62.7_{\pm 0.3}$ | 86.24 |

Figure 3: Here the gates $G_1, G_2, G_3, G_4$ are generated by the feature network and are permuted as $G_{i_1}, G_{i_2}, G_{i_3}, G_{i_4}$ before applying to the value network. $C_1, C_2, C_3, C_4$ have 128 filters each. **Table I and II:** All columns (except the last) show the $\%$ test accuracy on CIFAR-10 and CIFAR-100, and $\%$ of DNN performance recovered by DLGN is in the last column. **Table I:** For each dataset, the top row has results for vanilla models without permutations (the results are averaged over 5 runs) and the bottom row has results of $4! - 1 = 23$ permutations (except the identity) for each model (the results are averaged over the 23 permutations).

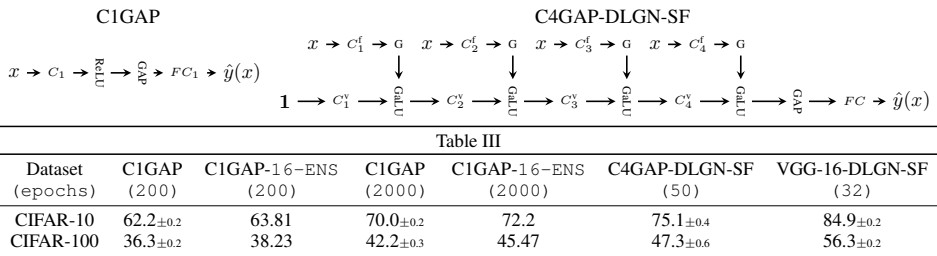

| | | Table III | | | |
|---|---|---|---|---|---|
| Dataset (epochs) | C1GAP (200) | C1GAP-16-ENS (200) | C1GAP (2000) | C1GAP-16-ENS (2000) | C4GAP-DLGN-SF (50) | VGG-16-DLGN-SF (32) |
| CIFAR-10 | $62.2_{\pm 0.2}$ | 63.81 | $70.0_{\pm 0.2}$ | 72.2 | $75.1_{\pm 0.4}$ | $84.9_{\pm 0.2}$ |
| CIFAR-100 | $36.3_{\pm 0.2}$ | 38.23 | $42.2_{\pm 0.3}$ | 45.47 | $47.3_{\pm 0.6}$ | $56.3_{\pm 0.2}$ |

Figure 4: C1GAP and C4GAP have width = 512 to make them comparable to VGG-16 whose maximum width is 512. The ensemble size is 16 to match the 16 layers of VGG-16. Note that C4GAP in Figure 3 has width=128. The last two columns show only DLGN-SF$(x, \mathbf{1})$. We observed the performance of DLGN-SF$(x, x)$ to be $\sim 2\%$ lesser and left it out from Table III for sake brevity. All results except for ENS are averaged over 5 runs.

## 3.3 INTERESTING FUTURE DIRECTIONS

Motivated by the success of DLGN, we further break it down into DLGN-*Shallow Features* (DLGN-SF), wherein, the feature network is a collection of shallow single matrix multiplications. We compare a shallow DNN with ReLU called C1GAP against DLGN-SF of C4GAP (see Figure 4) and VGG-16 (see Appendix). The results are in Figure 4, based on which we observe the following:

• **Power of depth in value network and lifting to dual space.** Both C1GAP and C4GAP-DLGN-SF were trained with identical batch size, optimiser and learning rate (chosen to be the best for C1GAP). The performance of C1GAP at 200 epochs is $\sim 10\%$ lower than that of C4GAP-DLGN-SF. After 2000 epochs of training and ensembling 16 such C1GAPs as C1GAP-16-ENS closes the gap within $\sim 3\%$ on C4GAP-DLGN-SF. Yet, a deeper architecture VGG-16-DLGN-SF is $\sim 10\%$ better than C1GAP-16-ENS. Note that both VGG-16-DLGN-SF and C1GAP-16-ENS have gates for 16 layers produced in a shallow manner. While in a C1GAP-16-ENS, '16' C1GAPs are ensembled, in VGG-16-DLGN-SF these gates for 16 layers are used as gating signals to turn 'on/off' the GaLUs laid depth-wise as 16 layers of the value network, which helps to lift the computations to the dual space. Thus, using the gates to **lift** (instead of ensembling) the computations to the dual space in the value network is playing a critical role, investigating which is an important future work.

• **Power of depth in feature network.** By comparing CIFAR-100 performance of VGG-16-DLGN-SF in Figure 4 and that of VGG-16-DLGN in Figure 3, we see $\sim 6\%$ improvement if we have a deep linear network instead of many shallow linear networks as the feature network. This implies depth helps even if the feature network is entirely linear, investigating which is an important future work.

• **DGN vs DLGN** In Table I and II of Figure 3, the difference between DGN and DNN is minimal (about $3\%$), however, the difference between DLGN and DNN is significantly large. Thus, it is important to understand the role of the ReLUs in the feature network of DGN. It is interesting to know whether this is simpler than understanding the DNN with ReLUs itself.

• **Is DLGN a Universal Spectral Approximator?** The value network realises the NPK which in general is an ensemble (assuming skip connections). The NPK is based on the gates whose pre-activations are generated linearly. It is interesting to ask whether the DLGN via its feature network learns the right linear transformations to extract the relevant spectral features (to tigger the gates) and via its value network learns the ensembling of kernels (based on gates) in a dataset dependent manner.

## 4 CONCLUSION

Entanglement of the non-linear and the linear operation in each layer of a DNN makes them uninterpretable. This paper proposed a novel DLGN which disentangled the computations in a DNN with ReLUs into two mathematically interpretable linearities, the 'primal' linearity from the input to the pre-activations that trigger the gates, and the 'dual' linearity in the path space. DLGN recovers more than $83.5\%$ of performance of state-of-the-art DNNs on CIFAR-10 and CIFAR-100. Based on this success of DLGN, the paper concluded by identifying several interesting future directions.

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

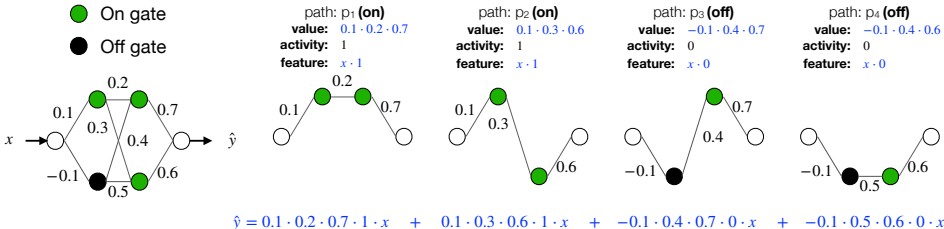

Figure 5: Illustration of Definition 2.1 and Proposition 2.1 in a toy network with 2 layers, 2 gates per layer and 4 paths. Paths $p_1$ and $p_2$ are 'on' and paths $p_3$ and $p_4$ are 'off'. The value, activity and feature of the individual paths are shown. $\hat{y}$ is the summation of the individual path contributions.

# A  FULLY CONNECTED

Here, we present the formal definition for the neural path features and neural path values for the fully connected case in Definition A.1. The layer-by-layer way of expressing the computation in a DNN of width '$w$' and depth '$d$' is given below.

| | | | |
|---|---|---|---|
| Input Layer | : | $z_{x,\Theta}(\cdot, 0)$ | $=$ | $x$ |
| Pre-Activation | : | $q_{x,\Theta}(i_{\text{out}}, l)$ | $=$ | $\sum_{i_{\text{in}}} \Theta(i_{\text{in}}, i_{\text{out}}, l) \cdot z_{x,\Theta}(i_{\text{in}}, l-1)$ |
| Gating | : | $G_{x,\Theta}(i_{\text{out}}, l)$ | $=$ | $\mathbf{1}_{\{q_{x,\Theta}(i_{\text{out}}, l) > 0\}}$ |
| Hidden Layer Output | : | $z_{x,\Theta}(i_{\text{out}}, l)$ | $=$ | $q_{x,\Theta}(i_{\text{out}}, l) \cdot G_{x,\Theta}(i_{\text{out}}, l)$ |
| Final Output | : | $\hat{y}_\Theta(x)$ | $=$ | $\sum_{i_{\text{in}}} \Theta(i_{\text{in}}, i_{\text{out}}, d) \cdot z_{x,\Theta}(i_{\text{in}}, d-1)$ |

Table 1: Information flow in a FC-DNN with ReLU. Here, '$q$'s are pre-activation inputs, '$z$'s are output of the hidden layers, '$G$'s are the gating values. $l \in [d-1]$ is the index of the layer, $i_{\text{out}}$ and $i_{\text{in}}$ are indices of nodes in the current and previous layer respectively.

**Notation A.1.** *Index maps identify the nodes through which a path $p$ passes. The ranges of index maps $\mathcal{I}_l^f, \mathcal{I}_l, l \in [d-1]$ are $[d_{in}]$ and $[w]$ respectively. $\mathcal{I}_d(p) = 1, \forall p \in [P^{fc}]$.*

**Definition A.1.** *Let $x \in \mathbb{R}^{d_{in}}$ be the input to the DNN. For this input,*

*(i) $A_\Theta(x, p) \triangleq \Pi_{l=1}^{d-1} G_{x,\Theta}(\mathcal{I}_l(p), l)$ is the activity of a path.*

*(ii) $\phi_\Theta(x) \triangleq \left( x(\mathcal{I}_0^f(p)) A_\Theta(x, p), p \in [P^{fc}] \right) \in \mathbb{R}^{P^{fc}}$ is the neural path feature (NPF).*

*(iii) $v_\Theta \triangleq \left( \Pi_{l=1}^{d} \Theta(\mathcal{I}_{l-1}(p), \mathcal{I}_l(p), l), p \in [P^{fc}] \right) \in \mathbb{R}^{P^{fc}}$ is the neural path value (NPV).*

# B  CONVOLUTION WITH GLOBAL AVERAGE POOLING

In this section, we define NPFs and NPV in the presence of convolution with pooling. This requires three key steps (i) treating pooling layers like gates/masks (see Definition B.2) (ii) bundling together the paths that share the same path value (due to weight sharing in convolutions, see Definition B.3), and (iii) re-defining the NPF and NPV for bundles (see Definition B.4). Weight sharing due to convolutions and pooling makes the NPK rotationally invariant Lemma B.1. We begin by describing the architecture.

**Architecture:** We consider (for sake of brevity) a 1-dimensional[1] convolutional neural network with circular convolutions, with $d_{\text{cv}}$ convolutional layers ($l = 1, \ldots, d_{\text{cv}}$), followed by a *global-average-pooling* layer ($l = d_{\text{cv}} + 1$) and $d_{\text{fc}}$ ($l = d_{\text{cv}} + 2, \ldots, d_{\text{cv}} + d_{\text{fc}} + 1$) fully connected layers. The convolutional window size is $w_{\text{cv}} < d_{\text{in}}$, the number of filters per convolutional layer as well as the width of the FC is $w$.

**Indexing:** Here $i_{\text{in}}/i_{\text{out}}$ are the indices (taking values in $[w]$) of the input/output filters. $i_{\text{cv}}$ denotes the indices of the convolutional window taking values in $[w_{\text{cv}}]$. $i_{\text{fout}}$ denotes the indices (taking values

---

[1]The results follow in a direct manner to any form of circular convolutions.

in $[d_{\text{in}}]$, the dimension of input features) of individual nodes in a given output filter. The weights of layers $l \in [d_{\text{cv}}]$ are denoted by $\Theta(i_{\text{cv}}, i_{\text{in}}, i_{\text{out}}, l)$ and for layers $l \in [d_{\text{fc}}] + d_{\text{cv}}$ are denoted by $\Theta(i_{\text{in}}, i_{\text{out}}, l)$. The pre-activations, gating and hidden unit outputs are denoted by $q_{x,\Theta}(i_{\text{fout}}, i_{\text{out}}, l)$, $G_{x,\Theta}(i_{\text{fout}}, i_{\text{out}}, l)$, and $z_{x,\Theta}(i_{\text{fout}}, i_{\text{out}}, l)$ for layers $l = 1, \ldots, d_{\text{cv}}$.

**Definition B.1** (Circular Convolution). *For $x \in \mathbb{R}^{d_{in}}$, $i \in [d_{in}]$ and $r \in \{0, \ldots, d_{in} - 1\}$, define :*

*(i) $i \oplus r = i + r$, for $i + r \leq d_{in}$ and $i \oplus r = i + r - d_{in}$, for $i + r > d_{in}$.*

*(ii) $rot(x, r)(i) = x(i \oplus r), i \in [d_{in}]$.*

*(iii) $q_{x,\Theta}(i_{fout}, i_{out}, l) = \sum_{i_{cv}, i_{in}} \Theta(i_{cv}, i_{in}, i_{out}, l) \cdot z_{x,\Theta}(i_{fout} \oplus (i_{cv} - 1), i_{in}, l - 1)$.*

**Definition B.2** (Pooling). *Let $G_{x,\Theta}^{pool}(i_{fout}, i_{out}, d_{cv} + 1)$ denote the pooling mask, then we have*
$$z_{x,\Theta}(i_{out}, d_{cv} + 1) = \sum_{i_{fout}} z_{x,\Theta}(i_{fout}, i_{out}, d_{cv}) \cdot G_{x,\Theta}^{pool}(i_{fout}, i_{out}, d_{cv} + 1),$$
*where in the case of* global-average-pooling $G_{x,\Theta}^{pool}(i_{fout}, i_{out}, d_{cv} + 1) = \frac{1}{d_{in}}, \forall i_{out} \in [w], i_{fout} \in [d_{in}]$.

| Input Layer | : | $z_{x,\Theta}(\cdot, 1, 0)$ | $=$ | $x$ |

| Convolutional Layers, $l \in [d_{\text{cv}}]$ | | | | |
|---|---|---|---|---|
| Pre-Activation | : | $q_{x,\Theta}(i_{\text{fout}}, i_{\text{out}}, l)$ | $=$ | $\sum_{i_{\text{cv}}, i_{\text{in}}} \Theta(i_{\text{cv}}, i_{\text{in}}, i_{\text{out}}, l) \cdot z_{x,\Theta}(i_{\text{fout}} \oplus (i_{\text{cv}} - 1), i_{\text{in}}, l - 1)$ |
| Gating Values | : | $G_{x,\Theta}(i_{\text{fout}}, i_{\text{out}}, l)$ | $=$ | $\mathbf{1}_{\{q_{x,\Theta}(i_{\text{fout}}, i_{\text{out}}, l) > 0\}}$ |
| Hidden Unit Output | : | $z_{x,\Theta}(i_{\text{fout}}, i_{\text{out}}, l)$ | $=$ | $q_{x,\Theta}(i_{\text{fout}}, i_{\text{out}}, l) \cdot G_{x,\Theta}(i_{\text{fout}}, i_{\text{out}}, l)$ |

| GAP Layer, $l = d_{\text{cv}} + 1$ | | | | |
|---|---|---|---|---|
| Hidden Unit Output | : | $z_{x,\Theta}(i_{\text{out}}, d_{\text{cv}} + 1)$ | $=$ | $\sum_{i_{\text{fout}}} z_{x,\Theta}(i_{\text{fout}}, i_{\text{out}}, d_{\text{cv}}) \cdot G_{x,\Theta}^{\text{pool}}(i_{\text{fout}}, i_{\text{out}}, d_{\text{cv}} + 1)$ |

| Fully Connected Layers, $l \in [d_{\text{fc}}] + (d_{\text{cv}} + 1)$ | | | | |
|---|---|---|---|---|
| Pre-Activation | : | $q_{x,\Theta}(i_{\text{out}}, l)$ | $=$ | $\sum_{i_{\text{in}}} \Theta(i_{\text{in}}, i_{\text{out}}, l) \cdot z_{x,\Theta}(i_{\text{in}}, l - 1)$ |
| Gating Values | : | $G_{x,\Theta}(i_{\text{out}}, l)$ | $=$ | $\mathbf{1}_{\{(q_{x,\Theta}(i_{\text{out}}, l)) > 0\}}$ |
| Hidden Unit Output | : | $z_{x,\Theta}(i_{\text{out}}, l)$ | $=$ | $q_{x,\Theta}(i_{\text{out}}, l) \cdot G_{x,\Theta}(i_{\text{out}}, l)$ |
| Final Output | : | $\hat{y}_{\Theta}(x)$ | $=$ | $\sum_{i_{\text{in}}} \Theta(i_{\text{in}}, i_{\text{out}}, d) \cdot z_{x,\Theta}(i_{\text{in}}, d - 1)$ |

Table 2: Shows the information flow in the convolutional architecture described at the beginning of Appendix B.

## B.1 NEURAL PATH FEATURES, NEURAL PATH VALUE

**Proposition B.1.** *The total number of paths in a CNN is given by $P^{cnn} = d_{in}(w_{cv}w)^{d_{cv}}w^{(d_{fc}-1)}$.*

**Notation B.1** (Index Maps). *The ranges of index maps $\mathcal{I}_l^f$, $\mathcal{I}_l^{cv}$, $\mathcal{I}_l$ are $[d_{in}]$, $[w_{cv}]$ and $[w]$ respectively.*

**Definition B.3** (Bundle Paths of Sharing Weights). *Let $\hat{P}^{cnn} = \frac{P^{cnn}}{d_{in}}$, and $\{B_1, \ldots, B_{\hat{P}^{cnn}}\}$ be a collection of sets such that $\forall i, j \in [\hat{P}^{cnn}], i \neq j$ we have $B_i \cap B_j = \emptyset$ and $\cup_{i=1}^{\hat{P}^{cnn}} B_i = [P^{cnn}]$. Further, if paths $p, p' \in B_i$, then $\mathcal{I}_l^{cv}(p) = \mathcal{I}_l^{cv}(p'), \forall l = 1, \ldots, d_{cv}$ and $\mathcal{I}_l(p) = \mathcal{I}_l(p'), \forall l = 0, \ldots, d_{cv}$.*

**Proposition B.2.** *There are exactly $d_{in}$ paths in a bundle.*

**Definition B.4.** *Let $x \in \mathbb{R}^{d_{in}}$ be the input to the CNN. For this input,*
$$A_{\Theta}(x, p) \triangleq \left( \Pi_{l=1}^{d_{cv}+1} G_{x,\Theta}(\mathcal{I}_l^f(p), \mathcal{I}_l(p), l) \right) \cdot \left( \Pi_{l=d_{cv}+2}^{d_{cv}+d_{fc}+1} G_{x,\Theta}(\mathcal{I}_l(p), l) \right)$$
$$\phi_{x,\Theta}(\hat{p}) \triangleq \sum_{\hat{p} \in B_{\hat{p}}} x(\mathcal{I}_0^f(p)) A_{\Theta}(x, p)$$
$$v_{\Theta}(B_{\hat{p}}) \triangleq \left( \Pi_{l=1}^{d_{cv}} \Theta(\mathcal{I}_l^{cv}(p), \mathcal{I}_{l-1}(p), \mathcal{I}_l(p), l) \right) \cdot \left( \Pi_{l=d_{cv}+2}^{d_{cv}+d_{fc}+1} \Theta(\mathcal{I}_{l-1}(p), \mathcal{I}_l(p), l) \right)$$

| NPF | $\phi_{x,\Theta} \triangleq (\phi_{x,\Theta}(B_{\hat{p}}), \hat{p} \in [\hat{P}^{cnn}]) \in \mathbb{R}^{\hat{P}^{cnn}}$ |
|---|---|
| NPV | $v_{\Theta} \triangleq (v_{\Theta}(B_{\hat{p}}), \hat{p} \in [\hat{P}^{cnn}]) \in \mathbb{R}^{\hat{P}^{cnn}}$ |

## B.2 ROTATIONAL INVARIANT KERNEL

**Lemma B.1.**

$$NPK_\Theta^{CONV}(x, x') = \sum_{r=0}^{d_{in}-1} \langle x, rot(x', r) \rangle_{\textbf{\textit{overlap}}_\Theta(\cdot, x, rot(x', r))}$$

$$= \sum_{r=0}^{d_{in}-1} \langle rot(x, r), x' \rangle_{\textbf{\textit{overlap}}_\Theta(\cdot, rot(x, r), x')}$$

*Proof.* For the CNN architecture considered in this paper, each bundle has exactly $d_{in}$ number of paths, each one corresponding to a distinct input node. For a bundle $b_{\hat{p}}$, let $b_{\hat{p}}(i), i \in [d_{in}]$ denote the path starting from input node $i$.

$$\sum_{\hat{p} \in [\hat{P}]} \left( \sum_{i, i' \in [d_{in}]} x(i) x'(i') A_\Theta(x, b_{\hat{p}}(i)) A_\Theta(x', b_{\hat{p}}(i')) \right)$$

$$= \sum_{\hat{p} \in [\hat{P}]} \left( \sum_{i \in [d_{in}], i' = i \oplus r, r \in \{0, \ldots, d_{in}-1\}} x(i) x'(i \oplus r) A_\Theta(x, b_{\hat{p}}(i)) A_\Theta(x', b_{\hat{p}}(i \oplus r)) \right)$$

$$= \sum_{\hat{p} \in [\hat{P}]} \left( \sum_{i \in [d_{in}], r \in \{0, \ldots, d_{in}-1\}} x(i) rot(x', r)(i) A_\Theta(x, b_{\hat{p}}(i)) A_\Theta(rot(x', r), b_{\hat{p}}(i)) \right)$$

$$= \sum_{r=0}^{d_{in}-1} \left( \sum_{i \in [d_{in}]} x(i) rot(x', r)(i) \sum_{\hat{p} \in [\hat{P}]} A_\Theta(x, b_{\hat{p}}(i)) A_\Theta(rot(x', r), b_{\hat{p}}(i)) \right)$$

$$= \sum_{r=0}^{d_{in}-1} \left( \sum_{i \in [d_{in}]} x(i) rot(x', r)(i) \textbf{overlap}_\Theta(i, x, rot(x', r)) \right)$$

$$= \sum_{r=0}^{d_{in}-1} \langle x, rot(x', r) \rangle_{\textbf{overlap}_\Theta(\cdot, x, rot(x', r))}$$

$\square$

In what follows we re-state Theorem 3.2.

**Theorem B.1.** *Let* $\sigma_{cv} = \frac{c_{scale}}{\sqrt{w w_{cv}}}$ *for the convolutional layers and* $\sigma_{fc} = \frac{c_{scale}}{\sqrt{w}}$ *for FC layers. Under Assumption 2.1, as* $w \to \infty$, *with* $\beta_{cv} = \left( d_{cv} \sigma_{cv}^{2(d_{cv}-1)} \sigma_{fc}^{2d_{fc}} + d_{fc} \sigma_{cv}^{2d_{cv}} \sigma_{fc}^{2(d_{fc}-1)} \right)$ *we have:*

$$NTK_{\Theta_0^{DGN}}^{CONV} \to \quad \frac{\beta_{cv}}{d_{in}^2} \cdot NPK_{\Theta_0^f}^{CONV}$$

*Proof.* Follows from Theorem 5.1 in [13]. $\square$

## C RESIDUAL NETWORKS WITH SKIP CONNECTIONS

As a consequence of the skip connections, within the ResNet architecture there are $2^b$ sub-FC networks (see Definition 3.1). The total number of paths $P^{res}$ in the ResNet is equal to the summation of the paths in these $2^b$ sub-FC networks (see Proposition C.1). Now, The neural path features and the neural path value are $P^{res}$ dimensional quantities, obtained as the concatenation of the NPFs and NPV of the $2^b$ sub-FC networks.

**Proposition C.1.** *The total number of paths in the ResNet is* $P^{res} = d_{in} \cdot \sum_{i=0}^{b} \binom{b}{i} w^{(i+2)d_{blk}-1}$.

**Lemma C.1** (Sum of Product Kernel). *Let $NPK_{\Theta}^{RES}$ be the NPK of the ResNet, and $NPK_{\Theta}^{\mathcal{J}}$ be the NPK of the sub-FCNs within the ResNet obtained by ignoring those skip connections in the set $\mathcal{J}$. Then,*

$$NPK_{\Theta}^{RES} = \sum_{\mathcal{J} \in 2^{[b]}} NPK_{\Theta}^{\mathcal{J}}$$

*Proof.* Proof is complete by noting that the NPF of the ResNet is a concatenation of the NPFs of the $2^b$ distinct sub-FC-DNNs within the ResNet architecture. $\square$

We re-state Theorem 3.3

**Theorem C.1.** *Let $\sigma = \frac{c_{scale}}{\sqrt{w}}$. Under Assumption 2.1, as $w \to \infty$, for $\beta_{res}^{\mathcal{J}} = (|\mathcal{J}| + 2) \cdot d_{blk} \cdot \sigma^{2\left((|\mathcal{J}|+2)d_{blk}-1\right)}$,*

$$NTK_{\Theta_0^{DGN}}^{RES} \to \sum_{\mathcal{J} \in 2^{[b]}} \beta_{res}^{\mathcal{J}} NPK_{\Theta_0^f}^{\mathcal{J}}$$

*Proof.* Follows from Theorem 5.1 in [13]. $\square$

## D NUMERICAL EXPERIMENTS: SETUP DETAILS

We now list the details related to the numerical experiments which have been left out in the main body of the paper.

• Computational Resource. The numerical experiments were run in Nvidia-RTX 2080 TI GPUs and Tesla V100 GPUs.

• All the models in Table I of Figure 3 we used Adam (Kingma & Ba, 2014) with learning rate of $3 \times 10^{-4}$, and batch size of 32.

• We tried several values of $\beta$ in the range from 1 to 100, and found the range 4 to 10 to be suitable. We have chosen $\beta = 10$ throughout the experiments.

• In Section 3.2, the codes for experiments based on VGG-16 and Resnet-110 were refactored from following repository: "https://github.com/gahaalt/resnets-in-tensorflow2".

• For VGG-16-DLGN in Figure 3 and DLGN-SF in Figure 4, the max-pooling were replaced by *average* pooling so as to ensure that the feature network is entirely linear. For the comparison to be fair, we replaced the max pooling in VGG-16 reported in Figure 3 by *average* pooling. Batch normalisation layers were retained in VGG-16, VGG-16-DLGN and VGG-16-DLGN-SF (all three are shown in Figure 8).

• DLGN of Resnet-110 was derived from Resnet-110 in a similar manner.

• All the VGG-16, Resnet-110 (and their DGN/DLGN) models in Table II of Figure 3 we used *SGD* optimiser with momentum 0.9 and the following learning rate schedule (as suggested in "https://github.com/gahaalt/resnets-in-tensorflow2") : for iterations $[0, 400)$ learning rate was 0.01, for iterations $[400, 32000)$ the learning rate was 0.1, for iterations $[32000, 48000)$ the learning rate was 0.01, for iterations $[48000, 64000)$ the learning rate was 0.001. The batch size was 128. The models were trained till 32 epochs.

• The VGG-16-DLGN-SF in Table III of Figure 4 uses the same optimiser, batch size and learning rate schedule as the models in Table II of Figure 3 as explained in the previous point.

• For C1GAP and C4GAP in Table III of Figure 4, we used Adam (Kingma & Ba, 2014) with learning rate of $10^{-3}$, and batch size of 32. This learning rate is best among the set $\{10^{-1}, 10^{-2}, 10^{-3}, 3 \times 10^{-4}\}$ for C1GAP.

# E    MODELS USED IN FIGURE 3 AND FIGURE 4

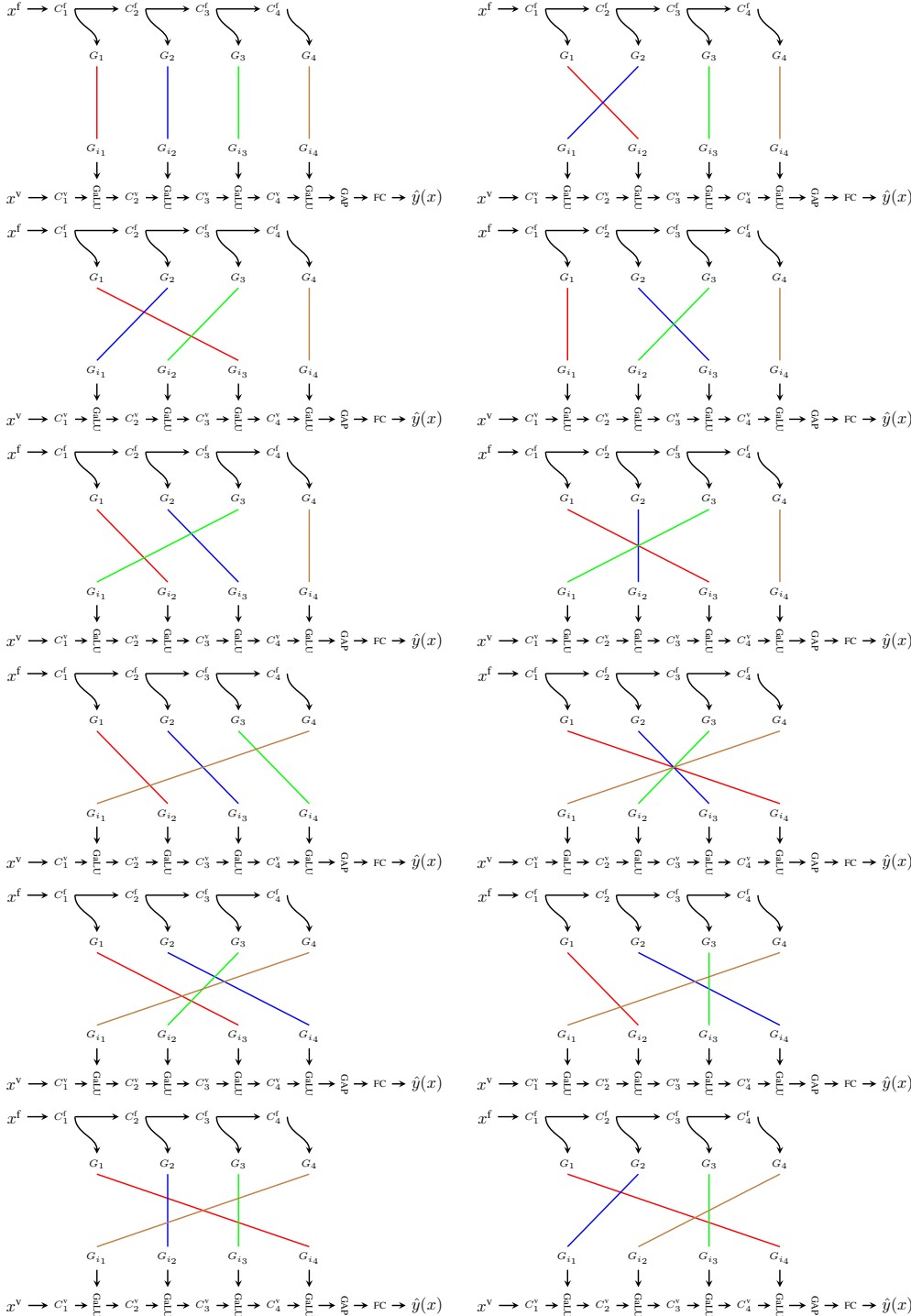

Figure 6: Shows the permutations $1 - 12$ C4GAP-DLGN in Table I of Figure 3. The top left is the identity permutation and is the vanilla model.

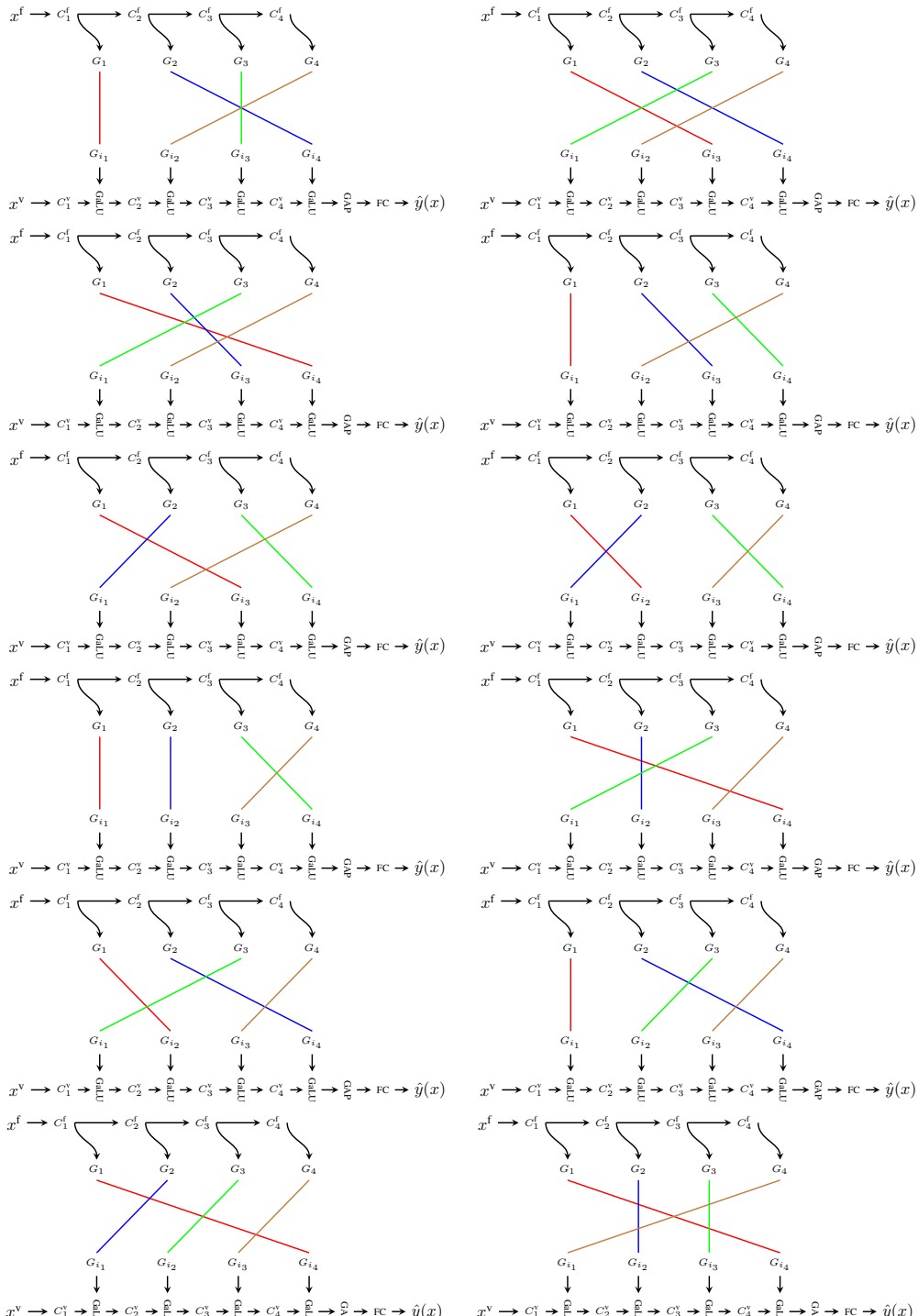

Figure 7: Shows the permutations $13 - 24$ C4GAP-DLGN in Table I of Figure 3.

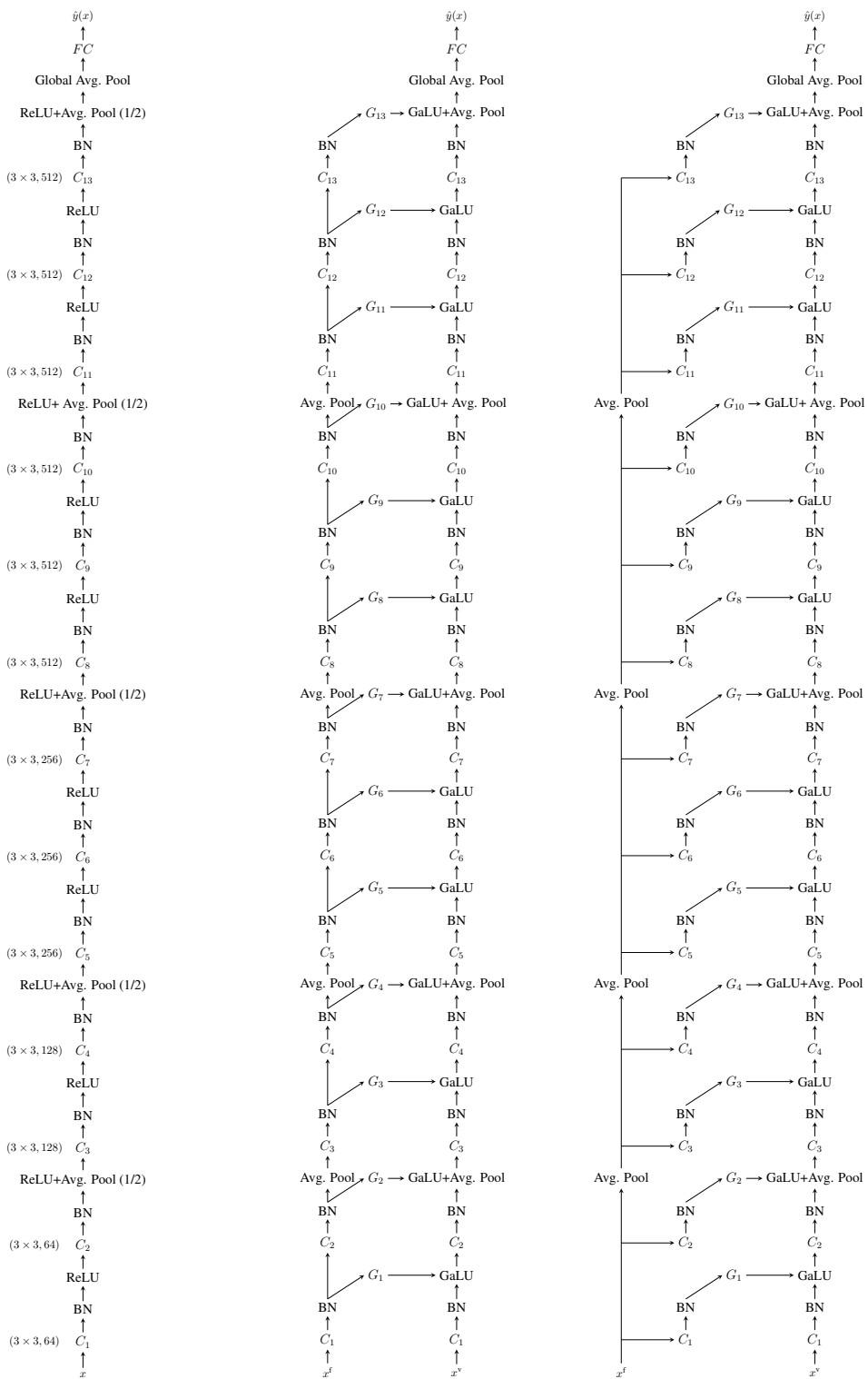

Figure 8: Shows VGG-16 (left), VGG-16-DLGN (middle), VGG-16-DLGN-SF(right).

## F  CONTINUAL LEARNING

In this section we compare DLGN and Gated Linear Network (GLN) of Veness et al. (2019). We highlight two closely related aspects of the GLN (as a result of the context based gating) namely (i) **context induced increased capacity(CIIC)** and (ii) **context induced sparse activity (CISA)**. These two aspects were present in the work by Veness et al. (2019), yet they were not highlighted because these two become apparent only from the equivalent *flattened computational graph* (see Appendix F.3) of the GLN (which was not present in Veness et al. (2019)). While Veness et al. (2019) attributed '**cosine similarity**' for the success of GLN in continual learning tasks, we show that cosine similarity itself is not sufficient and *context induced increased capacity* and *context induced sparse activity* are also necessary. We show via experiments that :

(i) GLN with CIIC and CISA performs well in continual learning.

(ii) GLN without CIIC and CISA performs poorly in continual learning.

(iii) DLGN-SF with increased capacity and sparse activity performs well in continual learning.

(iv) DLGN-SF without increased capacity and sparse activity performs poorly in continual learning.

**Continual Learning Task.** We compare the continual learning capabilities of the GLN and the DLGN-SF on the permuted MNIST dataset. We first generate 8 different permuted MNIST datasets and then use the same set of 8 tasks to study the continual learning capabilities of all the models.

### F.1  NON-COMPARABLE ASPECTS OF DLGN AND GLN

We first list and discuss the non-comparable aspects between the two works.

• **Training Algorithm:** The DLGN in our paper is a close cousin of the DNN with ReLUs and is trained by backpropagation. The GLN on the other hand is conceived and motivated from the *experts* setting and every node of the network is trained in a backpropagation free manner.

• **Learning in the Gates:** In our paper, the parameters of the feature network (whose layers trigger the gates) are learnt. In the GLN, the gating is fixed and not learnt. This is a very important difference because it was shown by Lakshminarayanan & Singh (2020) that the learning in the gates is a very important aspect of the DNNs with ReLUs.

• **Gating Mechanism:** In the case of DLGN, gating mechanism exists for fully connected, convolution, residual networks. In the GLN, there are no (equivalents of) convolutional/residual architectures, and as such we can say the GLN is equivalent to a fully connected architecture. Now, even in the case of fully connected architecture, the gating between DLGN and GLN differ quite a bit. In the case of DLGN, each unit is controlled by a gate which corresponds to a hyperplane. In the case of GLN, for each unit, the number of hyperplanes is equal to the so called *context-dimension*, and the binary encoding of these hyperplanes is mapped into one of $2^{\text{context-dimension}}$ contexts which is activated/selected.

### F.2  COMPARABLE ASPECTS OF DLGN AND GLN

Despite the aforementioned differences, DLGN and the GLN share the following commonalities (1) separate gating, (2) both models are *data dependent linear networks*. For the purpose of comparison to be close enough, let us consider fully connected DLGN-SF (shallow features) whose feature network is initialised at random and not trained. This way the DLGN-SF is similar to the GLN in that (a) the gates are fixed and (b) gates are generated via random hyperplanes. We now look at the data dependent linearity part by comparing the computational graphs of the DLGN-SF and GLN.

### F.3  FLATTING THE COMPUTATIONAL GRAPH OF GLN AND COMPARING IT WITH DLGN-SF

We now compare the computational graphs of GLN and the value network of DLGN-SF in Figure 9. For the sake of lucidity, in the Figure 9 as well in the discussion below, we have chosen input dimension to be 2, layers to be 2 and units per layer (i.e., width) to be 2. In Figure 9 we have omitted the gating part which will be discussed in the text below, along with other relevant details.

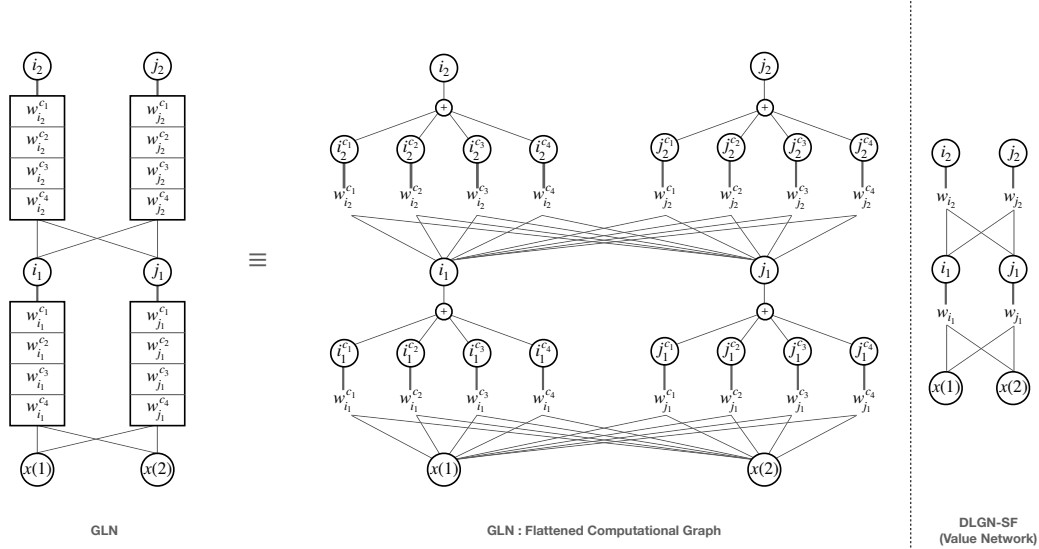

Figure 9: Shows a comparison of the gated linear network (GLN) and the value network of DLGN. For sake of clarity we have ommitted the details about gating which we have described in the text. Here $x = (x(1), x(2)) \in \mathbb{R}^2$ is the input. Each of the 'w's is a weight vector with 2 components. The bold lines connecting the units with their weight vectors denote their one-to-one correspondence. The 2 thin lines emanating from each $w$ are its 2 components (bold lines denote the entire weight vector).

**Left Most Diagram (Figure 9)** depicts (without gating) the GLN in a manner similar to Figure 1 of Veness et al. (2019). Here, $i_1$ and $j_1$ are the 2 units in layer 1 and $i_2$ and $j_2$ are the two units in layer 2. The 'w's are weight vectors in $\mathbb{R}^2$. The figure does not depict the 4 context maps $\mathcal{C}_{i_1}, \mathcal{C}_{j_1}, \mathcal{C}_{i_2}, \mathcal{C}_{j_2}$, where each $\mathcal{C}. : x \to \{c_1, c_2, c_3, c_4\}$. For example, if $C_{j_1}(x) = c_2$ then unit $j_1$ uses weight $w_{j_1}^{c_2}$, and its output will be $w_{j_1}^{c_2} x$.

**Middle Diagram (Figure 9)** is a *flattened* version of the GLN in the left most diagram. The aim of flattening is to depict the diagram in the left in the form of a standard computation graph. Flattening is achieved by expanding the units $i_1, j_1, i_2, j_2$ context wise into corresponding sub-units $i_1^{c_k}, j_1^{c_k}, i_2^{c_k}, j_2^{c_k}, k = 1, 2, 3, 4$. The function of the context maps can be achieved by *one-hot encoded* gating signals (this is not shown in Figure 9), $G_{i_1^{c_k}}(x), G_{j_1^{c_k}}(x), G_{i_2^{c_k}}(x), G_{j_2^{c_k}}(x), k = 1, 2, 3, 4$. For instance, if $\mathcal{C}_{i_2}(x) = c_3$, then we will have $G_{i_2^{c_3}}(x) = 1$ and $G_{i_2^{c_k}}(x) = 0, \forall k \neq 3$. The output of unit $i_1^{c_2}$ is $G_{i_1^{c_2}}(x) \cdot w_{i_1}^{c_2} x$. The output of unit $i_1$ is the summation of the outputs of $i_1^{c_1}, i_1^{c_2}, i_1^{c_3}, i_1^{c_4}$.

**Right Most Diagram (Figure 9)** shows the value network of a DLGN. The gating signals $G_{i_1}, G_{j_1}, G_{i_2}, G_{j_2}$ have been left out in the diagram. The output of unit $i_1$ is $G_{i_1}(x) \cdot w_{i_1} x$.

## F.4 Context Induced Increase In Capacity and Sparse Activity

Two notable aspects about the GLN are the following:

• **Context Induced Increase In Capacity.** Note that due to the presence of contexts, after flattening, we can see that the width of the GLN gets multiplied by a factor equal to no.of.contexts = $2^{\text{context-dimension}}$. In Figure 9, the width of the GLN is 2 (in the left more diagram) and after flattening it becomes $2 \times 2^2$ (in the middle diagram). The number of parameters in a GLN is equal to the number of parameters of the DLGN-SF times the same factor no.of.contexts = $2^{\text{context-dimension}}$, i.e., say in the first layer, instead of $w_{i_1}$ and $w_{j_1}$ in the right most diagram, in the middle diagram we have $w_{i_1}^{c_k}, w_{j_1}^{c_k}, k = 1, 2, 3, 4$.

• **Context Induced Sparse Activity:** Eventhough the flattened width of the GLN gets multiplied by the factor equal to no.of.contexts $= 2^{\text{context-dimension}}$, in each of the flattened layers only $\frac{1}{\text{no.of.context}}$ units are active for any given input.

### F.5 CONTINUAL LEARNING IN GLN WITH AND WITHOUT CIIC AND CISA

The main reason for the success of GLN in continual learning (in permuted MNIST task) that is highlighted by Veness et al. (2019) is "inputs close in terms of **cosine similarity** will give rise to similar data dependent weight matrices. Since each task-specific cluster of examples is far from each other in signature space, the amount of interference between tasks is significantly reduced, with the gating essentially acting as an implicit weight hashing mechanism". However, we show in the experiments below that **cosine similarity is not sufficient, context induced increased capacity and context induced sparse activity are also necessary.**

**Experimental Setup.** We trained two GLN models namely GLNM-CD4 and GLNM-CD1 each of which have 10 one-vs-all GLNs each with 4 layers with 128 units in the first 3 layers, and one unit in the final layer which serves as the output unit. The context dimension of GLNM-CD4 is equal to 4 and the context dimension of GLNM-CD1 is equal to 1. The code for the GLN (GLN training) is from Basu & Kuhnle (2020). While it is mentioned in Veness et al. (2019) that a 98% accuracy is achieved (on a single MNIST taksk), we obtain only 94.32%. There are two reasons for this (i) in order to keep the comparison between GLN and DLGN-SF fair, we did not *de-skew* the dataset (which is the default option while using the GLN in Basu & Kuhnle (2020)), and if skewing is added the accuracy improves to about 96.5% and (ii) we also confirmed with one of the authors of Veness et al. (2019) that 128 units might not be sufficient to achieve 98%. However, since our aim here is to understand the trend in continual learning we accept 94.32% as a reasonable ballpark performance. We now compare GLNM-CD4 and GLNM-CD1 in terms of CIIC and CISA.

**CIIC (GLNM-CD4 vs GLNM-CD1).** In GLNM-CD4, the flattened width is $128 \times 16$. In GLNM-CD1, the flattened width is $128 \times 2$.

**CISA (GLNM-CD4 vs GLNM-CD1).** GLNM-CD4 the only 128 out of the $128 \times 16$ units are active for any given input, i.e., only $\frac{1}{16}$ of the units are active. In GLNM-CD1, 128 out of the 256 units are active, i.e., $\frac{1}{2}$ of the units are active.

Thus we can see GLNM-CD4 is a GLN with CIIC and CISA, and GLNM-CD1 is a GLN without CIIC and CISA. The results are in Tables A, and B where the entry in $i^{th}$ row and $j^{th}$ column stands for the test accuracy in Task-$j$ while training Task-$i$. All the results are based on 3 runs.

| Table A: GLNM-CD4 | | | | | | | |
|---|---|---|---|---|---|---|---|
| | Test 1 | Test 2 | Test 3 | Test 4 | Test 5 | Test 6 | Test 7 | Test 8 |
| Train 1 | $94.32_{\pm 0.16}$ | - | - | - | - | - | - | - |
| Train 2 | $92.98_{\pm 0.21}$ | $94.27_{\pm 0.07}$ | - | - | - | - | - | - |
| Train 3 | $92.37_{\pm 0.15}$ | $93.63_{\pm 0.06}$ | $94.52_{\pm 0.07}$ | - | - | - | - | - |
| Train 4 | $91.41_{\pm 0.05}$ | $92.55_{\pm 0.07}$ | $93.15_{\pm 0.17}$ | $94.48_{\pm 0.1}$ | - | - | - | - |
| Train 5 | $89.53_{\pm 0.23}$ | $91.70_{\pm 0.27}$ | $92.33_{\pm 0.22}$ | $93.41_{\pm 0.21}$ | $94.31_{\pm 0.1}$ | - | - | - |
| Train 6 | $86.99_{\pm 1.02}$ | $90.94_{\pm 0.95}$ | $91.07_{\pm 0.68}$ | $91.84_{\pm 0.21}$ | $93.14_{\pm 0.5}$ | $94.20_{\pm 0.34}$ | - | - |
| Train 7 | $84.91_{\pm 0.73}$ | $89.94_{\pm 0.49}$ | $90.14_{\pm 0.26}$ | $90.41_{\pm 0.32}$ | $91.21_{\pm 0.5}$ | $93.06_{\pm 0.28}$ | $94.12_{\pm 0.29}$ | - |
| Train 8 | $82.86_{\pm 0.39}$ | $85.53_{\pm 1.27}$ | $88.22_{\pm 0.97}$ | $89.00_{\pm 0.5}$ | $90.49_{\pm 0.57}$ | $92.11_{\pm 0.5}$ | $92.6_{\pm 0.32}$ | $93.98_{\pm 0.31}$ |

| Table B: GLNMD-CD1 | | | | | | | |
|---|---|---|---|---|---|---|---|
| | Test 1 | Test 2 | Test 3 | Test 4 | Test 5 | Test 6 | Test 7 | Test 8 |
| Train1 | $90.28_{\pm 1.21}$ | - | - | - | - | - | - | - |
| Train2 | $81.89_{\pm 1.45}$ | $90.29_{\pm 1.09}$ | - | - | - | - | - | - |
| Train3 | $75.19_{\pm 4.45}$ | $84.18_{\pm 2.45}$ | $89.4_{\pm 0.24}$ | - | - | - | - | - |
| Train4 | $72.69_{\pm 0.71}$ | $80.95_{\pm 0.34}$ | $84.69_{\pm 1.57}$ | $91.3_{\pm 0.5}$ | - | - | - | - |
| Train5 | $61.02_{\pm 2.79}$ | $73.16_{\pm 2.53}$ | $76.06_{\pm 2.09}$ | $83.14_{\pm 0.74}$ | $91.42_{\pm 0.7}$ | - | - | - |
| Train6 | $53.22_{\pm 2.47}$ | $70.77_{\pm 2.65}$ | $73.63_{\pm 1.01}$ | $68.56_{\pm 0.52}$ | $84.92_{\pm 0.68}$ | $91.66_{\pm 0.36}$ | - | - |
| Train7 | $49.21_{\pm 4.6}$ | $61.20_{\pm 1.99}$ | $64.35_{\pm 1.14}$ | $72.85_{\pm 4.11}$ | $79.13_{\pm 2.18}$ | $83.25_{\pm 0.91}$ | $91.29_{\pm 0.29}$ | - |
| Train8 | $42.49_{\pm 7.31}$ | $49.30_{\pm 3.02}$ | $61.62_{\pm 2.97}$ | $59.35_{\pm 3.19}$ | $73.58_{\pm 5.2}$ | $78.97_{\pm 6.56}$ | $81.87_{\pm 3.04}$ | $90.19_{\pm 2.31}$ |

**Observation.** From Tables A and B, we can observe that the GLNM-CD4 with CIIC and CISA performs well and GLNM-CD1 without CIIC and CISA performs poorly, from which we can conclude

that *cosine similarity is not sufficient, context induced increased capacity and context induced sparse activity are also necessary.*

### F.6 Continual Learning in DLGN-SF with and without capacity and sparse activity

**Cosine Similarity holds in DLGN-SF.** Even in DLGN-SF, the gates are nothing but hyperplanes and similar inputs should trigger similar gates and activate similar data dependent linear networks. Therefore cosine similarity holds true for DLGN-SF as well.

**Increasing Capacity in DLGN-SF.** In order to make capacity of DLGN-SF comparable to that of GLN with $w_{\text{GLN}}$ and a given context-dimension, we let the width of the DLGN-SF to be $w_{\text{DLGN-SF}} = w_{\text{GLN}} \times 2^{\text{context-dimension}}$. Thus we consider two models namely DLGN-SF-128x16 which is comparable to GLNM-CD4 and DLGN-SF-128x2 which is comparable to GLNM-CD1. DLGN-SF-128x16 has 10 one-vs-all DLGN-SFs, each with 4 layers with $128 \times 16$ units in the first three layers, and one unit in the last layer which serves as the output unit. DLGN-SF-128x2 has 10 one-vs-all DLGN-SFs, each with 4 layers with $128 \times 2$ units in the first three layers, and one unit in the last layer which serves as the output unit.

**Inducing Sparse Activity in DLGN-SF.** The gates in a DLGN-SF are triggered by the feature network which contains $w_{\text{DLGN-SF}}$ hyperplanes. Since we are considering DLGN-SF with randomly initialised (from a symmetric distribution) and fixed feature network, it is reasonable to expect that $\sim \frac{w_{\text{DLGN-SF}}}{2}$ of the gates are active/on. Thus in order to induce sparse activity we sort the pre-activation signals of the gates and activate the gates corresponding to the top $\frac{1}{2^{\text{context-dimension}}}$ pre-activation signals. Thus, for the case of DLGN-SF-128x16 to ensure sparse activity we activate only the top $\frac{128 \times 16}{16} = 128$ gates. For the case of DLGN-SF-128x2, since we are comparing it with GLNM-CD1 whose context dimension is 1, we need $\sim \frac{\frac{w_{\text{DLGN-SF}}}{2}}{2}$ to be active, and hence we just let the pre-activations to directly trigger the gates.

**Experimental Setup.** We trained with SGD, batch size of 32 and learning rate of 0.32 (this is equivalent to a learning rate of $10^{-2}$ and batch size of 1). All the results are based on 3 runs. Table C shows DLGN-SF-128x16 with sparse activiy, and Table D shows DLGN-SF-128x16 without sparse activity. Table E shows DLGN-SF-128x2 which does not sparse activity.

Table C: DLGN-SF-128x16
Sparse Activity: Top 128 Gates Active

| | Test 1 | Test 2 | Test 3 | Test 4 | Test 5 | Test 6 | Test 7 | Test 8 |
|---|---|---|---|---|---|---|---|---|
| Train 1 | $96.08_{\pm 0.03}$ | - | - | - | - | - | - | - |
| Train 2 | $95.79_{\pm 0.25}$ | $96.52_{\pm 0.07}$ | - | - | - | - | - | - |
| Train 3 | $94.61_{\pm 0.27}$ | $96.45_{\pm 0.13}$ | $96.55_{\pm 0.24}$ | - | - | - | - | - |
| Train 4 | $93.64_{\pm 0.67}$ | $96.12_{\pm 0.18}$ | $96.45_{\pm 0.26}$ | $96.73_{\pm 0.09}$ | - | - | - | - |
| Train 5 | $91.43_{\pm 0.39}$ | $95.93_{\pm 0.15}$ | $96.32_{\pm 0.23}$ | $96.41_{\pm 0.09}$ | $96.78_{\pm 0.2}$ | - | - | - |
| Train 6 | $90.13_{\pm 1.00}$ | $95.66_{\pm 0.04}$ | $96.08_{\pm 0.25}$ | $95.94_{\pm 0.18}$ | $96.50_{\pm 0.19}$ | $96.95_{\pm 0.1}$ | - | - |
| Train 7 | $89.34_{\pm 1.18}$ | $95.03_{\pm 0.15}$ | $95.72_{\pm 0.09}$ | $95.73_{\pm 0.19}$ | $96.07_{\pm 0.21}$ | $96.70_{\pm 0.05}$ | $96.92_{\pm 0.03}$ | - |
| Train 8 | $88.87_{\pm 0.61}$ | $94.27_{\pm 0.30}$ | $95.46_{\pm 0.09}$ | $95.77_{\pm 0.15}$ | $95.98_{\pm 0.24}$ | $96.72_{\pm 0.04}$ | $96.57_{\pm 0.06}$ | $96.79_{\pm 0.14}$ |

Table D: DLGN-SF-128x16
No Sparse Activity: $\sim \frac{128 \times 16}{2}$ Gates are Active

| | Test 1 | Test 2 | Test 3 | Test 4 | Test 5 | Test 6 | Test 7 | Test 8 |
|---|---|---|---|---|---|---|---|---|
| Train 1 | $96.87_{\pm 0.02}$ | - | - | - | - | - | - | - |
| Train 2 | $96.07_{\pm 0.02}$ | $97.04_{\pm 0.08}$ | - | - | - | - | - | - |
| Train 3 | $90.60_{\pm 0.94}$ | $96.35_{\pm 0.35}$ | $97.06_{\pm 0.01}$ | - | - | - | - | - |
| Train 4 | $81.85_{\pm 0.60}$ | $91.89_{\pm 0.47}$ | $96.21_{\pm 0.40}$ | $96.97_{\pm 0.04}$ | - | - | - | - |
| Train 5 | $62.41_{\pm 4.27}$ | $84.61_{\pm 3.75}$ | $93.19_{\pm 0.41}$ | $95.72_{\pm 0.16}$ | $96.92_{\pm 0.01}$ | - | - | - |
| Train 6 | $59.02_{\pm 4.92}$ | $66.48_{\pm 2.49}$ | $80.74_{\pm 1.89}$ | $91.73_{\pm 1.28}$ | $96.45_{\pm 0.15}$ | $97.12_{\pm 0.08}$ | - | - |
| Train 7 | $46.57_{\pm 10.13}$ | $55.19_{\pm 2.36}$ | $66.82_{\pm 1.87}$ | $87.26_{\pm 1.54}$ | $94.36_{\pm 0.29}$ | $96.63_{\pm 0.10}$ | $97.2_{\pm 0.09}$ | - |
| Train 8 | $29.15_{\pm 8.43}$ | $54.17_{\pm 4.26}$ | $62.69_{\pm 3.32}$ | $76.67_{\pm 4.23}$ | $82.23_{\pm 3.96}$ | $90.83_{\pm 4.01}$ | $92.94_{\pm 4.33}$ | $93.96_{\pm 4.54}$ |

| Table E: DLGN-SF-128x2
No Sparse Activity: $\sim$ 128 Gates Active | | | | | | | |
|---|---|---|---|---|---|---|---|
| | Test 1 | Test 2 | Test 3 | Test 4 | Test 5 | Test 6 | Test 7 | Test 8 |
| Train1 | $96.29_{\pm 0.13}$ | - | - | - | - | - | - | - |
| Train2 | $94.85_{\pm 0.87}$ | $96.62_{\pm 0.03}$ | - | - | - | - | - | - |
| Train3 | $87.44_{\pm 0.89}$ | $94.96_{\pm 0.68}$ | $96.42_{\pm 0.2}$ | - | - | - | - | - |
| Train4 | $76.54_{\pm 1.47}$ | $87.62_{\pm 0.25}$ | $93.74_{\pm 0.62}$ | $96.59_{\pm 0.17}$ | - | - | - | - |
| Train5 | $60.69_{\pm 3.13}$ | $80.16_{\pm 2.18}$ | $87.55_{\pm 4.35}$ | $94.58_{\pm 0.13}$ | $96.44_{\pm 0.14}$ | - | - | - |
| Train6 | $40.89_{\pm 7.04}$ | $59.1_{\pm 3.65}$ | $71.62_{\pm 8.58}$ | $83.75_{\pm 1.13}$ | $94.46_{\pm 1.01}$ | $96.51_{\pm 0.2}$ | - | - |
| Train7 | $37.86_{\pm 2.74}$ | $46.3_{\pm 3.14}$ | $61.75_{\pm 5.84}$ | $76.75_{\pm 3.18}$ | $87.41_{\pm 1.39}$ | $94.5_{\pm 0.59}$ | $96.55_{\pm 0.17}$ | - |
| Train8 | $29.7_{\pm 4.05}$ | $42.08_{\pm 1.87}$ | $55.73_{\pm 6.33}$ | $63.63_{\pm 0.56}$ | $70.29_{\pm 3.85}$ | $82.15_{\pm 4.48}$ | $90.7_{\pm 1.42}$ | $96.36_{\pm 0.06}$ |

**Observation.** From Tables C with Tables D and E, we can observe that the DLGN-SF-128x16 with CIIC and CISA performs well and DLGN-SF-128x16 without CIIC and CISA as well as DLGN-SF-128x2 without CIIC and CISA perform poorly.

### F.7 IMPROVING CONTINUAL LEARNING IN DLGN-SF BY INTRODUCING SPARSE ACTIVITY

We now show that DLGN-SF-128x2 can be made to perform well if we introduce sparse activity. For this we sort the pre-activations to the gates and then trigger only the top 16 gates. The results are shown in Table F.

| Table F: DLGN-SF-128x2
Sparse Activity: Top 16 Gates Active | | | | | | | |
|---|---|---|---|---|---|---|---|
| | Test 1 | Test 2 | Test 3 | Test 4 | Test 5 | Test 6 | Test 7 | Test 8 |
| Train1 | $94.48_{\pm 0.16}$ | - | - | - | - | - | - | - |
| Train2 | $92.86_{\pm 0.41}$ | $95.18_{\pm 0.08}$ | - | - | - | - | - | - |
| Train3 | $91.61_{\pm 0.75}$ | $94.26_{\pm 0.48}$ | $95.56_{\pm 0.13}$ | - | - | - | - | - |
| Train4 | $91.35_{\pm 0.68}$ | $93.8_{\pm 0.48}$ | $95.14_{\pm 0.14}$ | $95.47_{\pm 0.14}$ | - | - | - | - |
| Train5 | $89.65_{\pm 0.98}$ | $93.41_{\pm 0.48}$ | $94.6_{\pm 0.15}$ | $94.87_{\pm 0.31}$ | $95.44_{\pm 0.14}$ | - | - | - |
| Train6 | $88.71_{\pm 0.69}$ | $92.5_{\pm 0.82}$ | $93.58_{\pm 0.14}$ | $94.02_{\pm 0.3}$ | $94.92_{\pm 0.15}$ | $95.54_{\pm 0.05}$ | - | - |
| Train7 | $88.24_{\pm 0.59}$ | $90.9_{\pm 0.77}$ | $92.82_{\pm 0.28}$ | $93.66_{\pm 0.34}$ | $94.41_{\pm 0.05}$ | $95.01_{\pm 0.05}$ | $95.33_{\pm 0.07}$ | - |
| Train8 | $87.54_{\pm 0.63}$ | $88.84_{\pm 0.69}$ | $92.41_{\pm 0.33}$ | $93.57_{\pm 0.28}$ | $94.03_{\pm 0.13}$ | $94.83_{\pm 0.19}$ | $94.65_{\pm 0.29}$ | $95.68_{\pm 0.11}$ |

**Observation** The difference between Table E and Table F is that the model in Table F has sparse activity, and we observe that it sparse activity ensures good performance.

### F.8 CONCLUSION

We conclude this section on continual learning by saying DLGN-SF with increased capacity and sparse activity performs well in continual learning. A comparison of the continual learning capabilities of GLN and DLGN to find out which is the optimal model out of the two will require a more thorough study which we defer to future work. Further, to perform well in continual learning, the models should be capable of learning many tasks, which naturally requires the models to have more capacity. In the models that perform well above, we note that capacity has been built in two ways (i) via the 10 models due to 'one-vs-all' and (ii) the context in the GLN increases the capacity by a factor equal to no.of.contexts, and in the DLGN-SF we ensure capacity increase by increasing the width. Sparsity helps to ensure that the different parts of networks are responsible for different contexts and they do not interfere with one another. High capacity and interference minimisation have also been mentioned in Cheung et al. (2019) in the context of continual learning.

