# OpenReview forum: "Disentangling deep neural networks with rectified linear units using duality"
_ICLR.cc/2022/Conference — ICLR 2022 Submitted_

### Official Review · Reviewer_pMWb · 2021-10-25

**Correctness:** 4
**Technical Novelty And Significance:** 3
**Empirical Novelty And Significance:** 3
**Recommendation:** 6
**Confidence:** 3

**Main Review:**

Pros:
1. It is a good idea to adopt deep linearly gated networks (DLGN) for interpreting DNNs with ReLU
2. This work also offers some theoretical analysis to better understand the proposed method.
3. Compared to the prior work on dual view, this paper presents more insights and new results for convolutional networks with global average and ResNet

Cons:
1. Did not conduct experiments on complex tasks, such as ImageNet and CelebA. I am curious about what experimental results they will be.
2. This work did not discuss the impact of $\beta$ in the soft gate on the experimental results
3. It is hard for the audience without background to understand some theorems.

**Summary Of The Paper:**

This paper proposed deep linearly gated networks (DLGN) for interpreting DNNs with ReLU activations based on a dual view. The proposed framework is able to completely disentangle the ‘gating network’ and the ‘weight network’. Finally, the experiment results demonstrate that DLGN can achieve good performance for classification on two benchmark datasets compared to the existing DNNs.

**Summary Of The Review:**

The idea of using deep linearly gated networks (DLGN) with dual view looks novel and interesting. I think this paper should be above the acceptance threshold.

---

> ### Author Response · Authors · 2021-11-17
> **Response to Reviewer pMWb**
>
> Thanks for the review.
>
> 1. CIFAR-10 is the standard benchmark for recent works on NTK Lee et al. (2020); Arora et al. (2019); Novak et al. (2018); Lee et al. (2018) (theoretical as well empirical). We have gone one step ahead by presenting results on CIFAR-100 as well. Experimental results on large scale datasets such as ImageNet will be presented in the longer version of our paper.
>
> 2. We have played with $\beta$ and found that $\beta$ in the range of $4$ to $10$ seems to give good results. We will mention this in the paper.
>
> **References**
> Sanjeev Arora, Simon S Du, Wei Hu, Zhiyuan Li, Russ R Salakhutdinov, and Ruosong Wang. On exact computation with an infinitely wide neural net. In Advances in Neural Information Processing Systems, pp. 8139–8148, 2019.
>
> Jaehoon Lee, Yasaman Bahri, Roman Novak, Samuel S Schoenholz, Jeffrey Pennington, and Jascha Sohl-Dickstein. Deep neural networks as gaussian processes. In International Conference on Learning Representations, 2018.
>
> Jaehoon Lee, Samuel Schoenholz, Jeffrey Pennington, Ben Adlam, Lechao Xiao, Roman Novak, and Jascha Sohl-Dickstein. Finite versus infinite neural networks: an empirical study. Advances in Neural Information Processing Systems, 33, 2020.
>
> Roman Novak, Lechao Xiao, Yasaman Bahri, Jaehoon Lee, Greg Yang, Jiri Hron, Daniel A Abolafia, Jeffrey Pennington, and Jascha Sohl-dickstein. Bayesian deep convolutional networks with many channels are gaussian processes. In International Conference on Learning Representations, 2018.

---

### Official Review · Reviewer_hqwS · 2021-11-02

**Correctness:** 3
**Technical Novelty And Significance:** 3
**Empirical Novelty And Significance:** 3
**Recommendation:** 6
**Confidence:** 2

**Main Review:**

Strengths:
1. The view of this paper is interesting since the success of the DNN is its non-linear ability. And the paper proposed to reduce its non-linearity.
2. And both the theory and experiments are sufficient to support their claims.

Weakness:

Way too theory. It is a little bit hard to read for me.

**Summary Of The Paper:**

This paper deals with the entanglement in the DNN through two steps. First, replacing the rectified linear units (ReLU) in the traditional DNN with Deep Linearly Gated Network (DLGN). Second, demonstrate the weighted network is disentangled in the path space.

**Summary Of The Review:**

The problem of this paper deal with is important, and the results show the effectiveness of their proposed method.

---

> ### Author Response · Authors · 2021-11-17
> **Réponse to Reviewer hqwS**
>
>
> Thanks for the review. The focus of the paper is indeed theoretical aspects of DNNs (backed by extensive experiments). We believe that we have made the exposition as clear as possible and if there are any specific comments we will be happy to address them.

---

### Official Review · Reviewer_nByk · 2021-11-03

**Correctness:** 3
**Technical Novelty And Significance:** 2
**Empirical Novelty And Significance:** Not applicable
**Recommendation:** 6
**Confidence:** 3

**Main Review:**

## Strengths:
1. Disentangling neural networks in the path space is a really interesting perspective, which may give new insight to understanding neural networks.
2. By theoretical analysis, the paper proves some novel propositions (e.g., ResNets have an ensemble structure, NPK is rotationally invariant) which might be useful to understand deep learning models.

## Weakness:
1. Since the paper is largely based on the previous work DGN (Lakshmi-narayanan & Singh (2020)), it would be better to provide more details and formal statements about the prerequisites. Otherwise, it would be hard to follow for readers who are unfamiliar with DGN. (e.g., Definition 2.1 should be a more formal definition with mathematical symbols and formula.)
2. The paper claims that "the learning in the weighted network happens path-by-path" with experiments and theoretical analysis. However, the "input 1" experiment and the "layer permutation" experiment on DLGN only prove that **DLGN can** be learned path-by-path, and this conclusion doesn't necessarily hold for the general neural network.
3. The paper claims that "Destroying structure by permuting the layers", but actually, according to Figure 3 (C4GAP-DLGN), I think it's somewhat overclaimed. Let q_1, ..., q_4 denote the pre-activation which generate G_1, ..., G_4. And we have q_i = (C_i^f C_{i-1}^f ... C_1^f)(x^f). Since C_i^f C_{i-1}^f ... C_1^f is linear, we can merge (C_i^f C_{i-1}^f ... C_1^f) to a single weight matrix/convolution kernel, which is denoted as P_i. Then P_i is just a common linear operator, and different q_i has its own unique P_i. It's obvious that we can permute them and get similar performance. I don't think it can be called a "destroying structure".

## Questions:
1. In Table I, why is the performance of models whose layers are permutated slightly higher than the original models?
2. Can the linear network functions replace the non-linear network?  If not, why try to design the network in a linear manner?

**Summary Of The Paper:**

The paper extends the framework of the deep gated network to the deep linearly gated network. Based on this extension, the paper investigates the separate part of the network. By theoretical analysis and empirical experiments, the paper argues that the neural network is learned path-by-path instead of layer-by-layer. Also, it present that the neural path kernel has several interesting properties.

**Summary Of The Review:**

The paper gives an in-depth theoretical analysis of neural networks. But some of its claims are not well-supported or well explained. The writing of the paper is ok, but the prerequisites of the paper can be more comprehensive.

---

> ### Author Response · Authors · 2021-11-17
> **Can the linear network functions replace the non-linear network? If not, why try to design the network in a linear manner?**
>
> The ‘blackbox’-ness of deep neural networks is an important issue in machine learning (ML). Recent times have seen the emergence of two distinct schools of thought in how to go about addressing ‘blackbox’-ness namely (i) intepretable ML, wherein, non-interpretable models are to be avoided, and the models need to interpretable and completely white-box by design, (ii) explainable ML, which takes a slightly milder position, in that, it is okay to build non-interpretable models as long as one can obtain post-hoc explanations of decisions of such non-interpretable models by using a simpler interpretable model. Rudin (2019) makes a compelling case for why interpretable ML it is desirable
> over explainable ML.
> The nice thing about linear models is that they are mathematically interpretable. And the proposal in this paper is not to replace the non-linear network by a linear network. Note that our DLGN is indeed a non-linear network, in which only the gating network (i.e., feature network) is a deep linear network and the weight network is linear only in the dual variables and is still non-linear in its weights. Thus, the proposal is to only design the gating network to be linear to make it mathematically intereprable, while leaving the non-linearity in the weights network as such because the computations in the weights network is linear in the dual variables and hence mathematically interpretable via the NPK. Thus, the overall goal is to win in mathematical interpretability via linearity.
> The benefit of linearity is that standard linear algebraic tools can be deployed to further our theoretical as well as domain specific understanding (such as investigating the properties of the learnt filters using tools from image processing etc). The DLGN is thus an important candidate for further investigation, which could provide valuable insights into the inner workings of DNNs.
>
> **Reference**
> Cynthia Rudin. Stop explaining black box machine learning models for high stakes decisions and use interpretable models instead. Nature Machine Intelligence, 1(5):206–215, 2019.

---

> ### Author Response · Authors · 2021-11-17
> **In Table I, why is the performance of models whose layers are permutated slightly higher than the original models?**
>
>
> The reason for the marginal increase in performance of the permuted models is not known clearly. In the case of convolutional networks, while the NPK retains the rotational invariance property under permutations, the NPK is not identical across permutations. Thus, at this stage, our educated guess is that permuting the layers adds some kind of a regularising effect, understanding which will be part of future work.

---

> ### Author Response · Authors · 2021-11-17
> **Merging**
>
> $\textbf{Reviewer Comment:}$
> $\textbf{we can merge ($C_i^f C_{i-1}^f ... C_1^f$) to a single weight matrix/convolution kernel, which is denoted as $P_i$.}$$\textbf{ Then $P_i$ is just a common linear operator, and different $q_i$ has its own unique $P_i$.}$$\textbf{It's obvious that we can permute them and get similar performance}$
>
> While the above reduction is not valid for DGN, it is valid for DLGN. We have already investigated whether such merging will result in similar performance by comparing DLGN vs DLGN-SF (shallow features, i.e., each $q_i$ has  a single $P_i$). The interesting result as mentioned in the paper is that "By comparing CIFAR-100 performance of VGG-16-DLGN-SF in Figure 4 and that of VGG-16-DLGN in Figure 3, we see $∼ 6\%$ improvement". In short, such merging seems to degrade performance, a phenomena that is an interesting  future work.

---

> ### Author Response · Authors · 2021-11-17
> **"Destroying Strucuture" is over claimed ...don't think it can be called a "destroying structure"**
>
> We would like to stress that "Destroying structure by permuting the layers" is not over claimed as the reviewer mentions. We would like to point out to the reviewer that Figure 3 as well as Table I also contains C4GAP-DGN in which the reduction suggested by the reviewer for the case of C4GAP-DLGN will not apply, and indeed we are destroying the layer-by-layer structure.

---

> ### Author Response · Authors · 2021-11-17
> **DLGN can be learned path-by-path, and this conclusion doesn't necessarily hold for the general neural network**
>
> While we agree that "the learning in the weighted network happens path-by-path" is a claim specific to the value network (i.e., weight network) alone, we disagree that it is a weakness. To elaborate, from duality it follows that output of DNN with ReLUs can be expressed as inner product of two terms namely NPF and NPV. In this paper, we provide a fundamental insight that the NPV learning in the weight network happens path-by-path. What we are presenting is a fundamental insight on one of the two terms (i.e., NPV) in the output expression, and calling it a weakness is not justified.

---

> ### Author Response · Authors · 2021-11-17
> **Formal Definition**
>
> We have added formal definitions in the appendix of the recently updated version of the paper.

---

> ### Author Response · Authors · 2021-11-17
> **Response to Reviewer nByk**
>
> We thank the reviewer. Please find below our responses.

---

### Official Review · Reviewer_4UFG · 2021-11-03

**Correctness:** 4
**Technical Novelty And Significance:** 3
**Empirical Novelty And Significance:** 3
**Recommendation:** 5
**Confidence:** 4

**Main Review:**

* This work provides an alternative perspective on understanding and interpreting how deep networks work. In particular, the current study overcomes the limitations of NTK framework, by proposing NPK (Neural Path Kernel) for DNNs with gating and dual pathways.

* Beyond interpretbility, what is the functional role of gating? Is there any computational benefit of using a gating network? Recent work in gated linear networks [1,2] have provided functional examples where the usage of gating pathways gives rise to computational benefits such as superior performance in continual learning tasks [1] and implementation of local learning rules [2]. Perhaps the connection to these lines of work can be discussed more clearly.

* Does the separation of gating network and value network introduce twice as many parameters? Is there any way to reduce the number of parameters in gating network, and can this be used to explain why the performance of the DLGN is a bit worse than SOTA?

* Description in Figure 1 is a bit confusing and not self-contained. Can Fig 1 be more detailed? For example, illustrate "Gating Signal", with a small figure of the functional form of G, to illustrate GaLU. (I understand that it becomes clear in Figure 2, but perhaps it can be drawn in Figure 1 too)

* I’m not sure if the entanglement of linear and and non-linear operations of DNNs being uninterpretable is a strong motivation for developing a new (interpretable) class of networks, especially given many ongoing studies on interpretability methods that can be used directly on deep nonlinear networks. Perhaps, the paper could benefit from an additional clarification on the role of interpretable models (as opposed to interpretability methods on uninterpretable models).

* Concluding a paper with a meta-note that this paper concludes with the question if DLGN a universal spectral approximator makes the paper sound a bit like a long introduction to the next paper. Perhaps it should be stated as an interesting direction for future study. Admittedly, exploring the expressivity and capacity of DLGN is an interesting question in its own right and it's unclear why this is  emphasized only at the end of the paper.

References

[1] Veness, J., Lattimore, T., Bhoopchand, A., Budden, D., Mattern, C., Grabska-Barwinska, A., Toth, P., Schmitt, S. and Hutter, M., 2019. Gated linear networks. arXiv preprint arXiv:1910.01526.

[2] Clark, D.G., Abbott, L.F. and Chung, S., 2021. Credit Assignment Through Broadcasting a Global Error Vector. arXiv preprint arXiv:2106.04089.

**Summary Of The Paper:**

The authors of this paper proposes a DLGN (Deep Linear Gated Networks), a novel class of deep networks, inspired by a recent dual view where the computation in DNNs is broken into two parts: learning in the gates and learning in the weights. The DLGN disentangles the computations into 2 parts: (1) "primal" part between input and the pre-activations in the gating network, and (2) "dual" part in the weights network, conditioned on inputs and gates. DLGN’s performance recovers 83.5% of SOTA DNNs. This development may lead to more interpretable deep network models that are also highly performant.

**Summary Of The Review:**

I think this paper continues an interesting line of research. The goal and the motivation of the paper, and the connection to the existing prior work could be made more clear, but once those are addressed I am willing to change my score.

---

> ### Author Response · Authors · 2021-11-17
> **DLGN as universal spectral approximator**
>
> We agree with the reviewer, that "it should be stated as an interesting direction for future study". We have updated the same in the recently uploaded version of the paper.

---

> ### Author Response · Authors · 2021-11-17
> **Interpretablity is a goal worth pursuing**
>
>
> Rudin (2019) provides several compelling arguments for why it is important to build models that are interpretable by design as opposed to seeking post-hoc explanations for decisions of non-interpretable models (i.e., as the reviewer mentions interpretability methods for non-interpretable models). Rudin et al. (2021) discusses the grand challenges in interpretablity. The DLGN in our paper resonates with the following aspects discussed in Rudin (2019); Rudin et al. (2021).
>
> $\bullet$ **Faithfulness To Original Model**. Most times the simpler models that are used to explain the decision of the more complicated non-interpretable models do not mimic the computations, thereby the explanations are not faithful which is an issue. The DLGN in our paper does not suffer from this issue because, the DLGN is obtained by rearranging the computations of a DNN with ReLUs in a mathematically principled manner.
>
> $\bullet$ **Disentanglement of neural networks**. Rudin et al. (2021)  `DNNs are the quintessential “black box” because the computations within its hidden layers are typically inscrutable'. In our paper, we disentangle DNNs with ReLUs.
>
> $\bullet$ **Domain Specific Interpretations**. The fact that the feature network of a DLGN is linear makes it amenable to domain specific analysis such as understanding the properties of the learnt convolutional filters using standard image processing tools. While the focus of the current work is mathematical interpretability, we envision that this is also the first step towards visual/human level interpretability methods in which we believe that the linearity of the feature network can be leveraged in a critical way. However, this is a separate research direction (which include signal/image processing elements) in itself requiring considerable time and effort, and hence we enlist it as part of future work.
>
> We would like to reiterate (as mentioned in the paper) that "The DLGN is not an alternative architecture per se, but a disentanglement and an interpretable re-arrangement of the computations in a DNN with ReLUs". In addition to being a new class (interpretable) of models, we would also like to think  that DLGNs as close interpretable cousins of DNNs with ReLU, both models belonging to the larger class of models namely deep gated networks. The aim is to gain more insights into the inner workings of DNNs with ReLUs by studying DLGNs, and in this paper we have demonstrated that DLGNs are indeed worth studying because DLGN counterparts of state-of-the-art models recover more than $83\\%$ of the performance  on standard datasets.
>
>
> **References**
>
> Cynthia Rudin. Stop explaining black box machine learning models for high stakes decisions and use interpretable models instead. Nature Machine Intelligence, 1(5):206–215, 2019.
>
> Cynthia Rudin, Chaofan Chen, Zhi Chen, Haiyang Huang, Lesia Semenova, and Chudi Zhong. Interpretable machine learning: Fundamental principles and 10 grand challenges. arXiv preprint arXiv:2103.11251, 2021.

---

> ### Author Response · Authors · 2021-11-17
> **Is the DLGN worse than SOTA due to 2 times more parameters?**
>
> Yes, the separation of gating and value networks introduces twice as many parameter. We ran the DLGNs of VGG-16 on CIFAR-10 and CIFAR-100 by sharing the parameters of the value and the feature networks, and obtained the results in the table below, from which, we infer that the $2\times$ parameters in a DLGN is not the reason why they perform poorly compared to SOTA. Also, note that the DGN also has $2\times$ more parameters compared to the DNN, and yet in Tables I and II of the paper, we see that the DGN matches the DNN within $2-3\\%$. Thus (despite the $2\times$ more parameters) the reason why DLGNs perform poorly but DGNs don't is that in DGNs due to the presence of ReLU activations in the feature network, in each layer all outputs with negative components are dropped out (since they are made to $0$). Thus in each layer of DGN significant amount of outputs are dropped out which perhaps has a regularising effect/sparsifying effect and in DLGN the network being entirely linear there is no such drop out.
>
>
> |   With Sharing|   | | |Without Sharing | | |
> | ----|    ----| ----|  ----| ----|    ----| ----|
> ||Dataset| DLGN(x,**1**)|DLGN(x,x)| | DLGN(x,**1**)|DLGN(x,x)|
> ||CIFAR-10| $86.5\tiny{\pm0.5}$|$86.9\tiny{\pm0.3}$||$87.0\tiny{\pm0.2}$|$87.0\tiny{\pm0.1}$|
> ||CIFAR-100|$60.73\tiny{\pm0.4}$ |$60.58\tiny{\pm0.3}$||$61.5\tiny{\pm0.1}$| $61.5\tiny{\pm0.2}$|
>
> **Note: The numbers for the without sharing case replicated from Tables I and II in the paper.**

---

> ### Author Response · Authors · 2021-11-17
> **Functional Role of Gating and Connection to Clark et al. (2021)**
>
>
> The connection between our paper and that of Clark et al. (2021) is through the dual view. To elaborate, Clark et al. (2021) deal with Vectorized Non-Negative Networks (VNN) whose special case (for the case when vector dimension equals 1) is a DNN with ReLUs with non-negative weights past first layer. As we understand, the key aspect of VNNs is the non-negativity of the weights past the first layer, and the non-negativity of the derivative of the activation function (which holds for Heaviside step function used in their paper), which enable a global error vector broadcasting (GEVB) rule, a non-backprogagation rule. We would like to point out that in VNN, the gating is not separate (however the authors mention in the context of vectorisation unfolding over time that gating can be made separate). We use dual view to disentangle DNNs with ReLUs, whereas, Clark et al. (2021) use the definitions of neural path activity and neural path value to show that the GEVB rule is sign aligned with the gradient of the VNNs. In particular, it is assumed that "for all training examples, each hidden unit has at least one active path with nonzero value connecting it to the output unit" (see page 17 of Clark et al. (2021)). Thus, the Clark et al. (2021) use the dual view to justify the GEVB rule, we use the dual view to in a much more fundamental way by disentangling the computations is a DNN with ReLUs and reagrraging them in the form of DLGN to improve mathematical interepretability.
>
> **Reference**
>
> David G Clark, LF Abbott, and SueYeon Chung. Credit assignment through broadcasting a global error vector. arXiv preprint arXiv:2106.04089, 2021.

---

> ### Author Response · Authors · 2021-11-17
> **Functional Role of Gating and Connection to Veness et al. (2019)**
>
> We have added a detailed discussion at the end of the appendix in the recently updated version of our paper. We mention the important points here.
>
> $\bullet$ **Non-Comparable Aspects**: There are many non-comparable aspects between the DLGN in our work and the Gated Linear Network (GLN) of Veness et al. (2019) such as (i) **training**: we use backpropagation to train DLGN vs Veness et al. (2019) propose a backpropagation free algorithm, (ii)  **learning in the gates**: gates are learnable in our paper vs gates are fixed and random in Veness et al. (2019), and (iii) **gating mechanism**: more importantly Veness et al. (2019) presents only a fully connected GLN and convolutional/residual equivalents of GLNs are yet to be explored vs we have presented DLGN counterparts of state-of-the-art convolutional and residual models such as VGG16 and ResNet110.
>
>
>
> $\bullet$ **Comparable Aspects**: Despite non-comparable aspects, the DLGN and the GLN share the following commonalities (i) separate gating, (ii) both models are *data dependent linear networks*.
>
> $\bullet$ **Cosine Similarity (GLN and DLGN)**: The **success of GLN in continual learning** is attributed to the cosine similarity property ``inputs close in terms of cosine similarity will give rise to similar data dependent weight matrices. Since each task-specific cluster of examples is far from each other in signature space, the amount of interference between tasks is significantly reduced, with the gating essentially acting as an implicit weight hashing mechanism". Since in DLGNs, the gates are generated by hyperplanes, **cosine similarity property holds true in the case of DLGN as well**.
>
>
> $\bullet$ **Context Induced Increased Capacity (CIIC) and Context Induced Sparse Activity (CISA)**: We highlight two key aspects of the GLNs namely (i) context induced increased capacity: the effective width and the number of parameters gets increased by a factor equal to the number of contexts, (ii) context induced sparse activity: eventhough the width is increased, since only one context is chosen for each unit for a given input, only a sparse part of the network is active. These two aspects were present in the work by Veness et al. (2019), yet they were not highlighted because these two become apparent only from the equivalent flattened computational graph of the GLN (which was not present in their work).
>
> $\bullet$ **Main Results**: While Veness et al. (2019) attributed **cosine similarity** for the success of GLN in continual learning tasks, we show that **cosine similarity itself is not sufficient and context induced increased capacity and context induced sparse activity are also necessary** We show via experiments that (added in the appendix of the recently updated version of the paper)  :
>
> (i) GLN with CIIC and CISA performs well in continual learning.
>
> (ii) GLN without CIIC and CISA performs poorly in continual learning.
>
> (iii) DLGN-SF with increased capacity and sparse activity performs well in continual learning.
>
> (iv) DLGN-SF without increased capacity and sparse activity performs poorly in continual learning.
>
> $\bullet$ **Conclusion**: We conclude that **DLGN-SF with increased capacity and sparse activity performs well in continual learning**. A comparison of the continual learning capabilities of GLN and DLGN to find out which is the optimal model out of the two will require a more thorough study which we defer to future work.
>
> **References**
> Joel Veness, Tor Lattimore, Avishkar Bhoopchand, David Budden, Christopher Mattern, Agnieszka Grabska-Barwinska, Peter Toth, Simon Schmitt, and Marcus Hutter. Gated linear networks. arXiv preprint arXiv:1910.01526, 2019.
>
> Anindya Basu and Alexander Kuhnle. PyGLN: Gated Linear Network implementations for NumPy, PyTorch, TensorFlow and JAX, 2020. URL https://github.com/aiwabdn/pygln.

---

> ### Author Response · Authors · 2021-11-17
> **Response to Reviewer 4UFG**
>
> We thank the reviewer. Please find below our responses.

---

### Decision · Program_Chairs · 2022-01-20

**Decision:**

Reject

**Comment:**

The paper examines a sum-over-paths representation of ReLU networks, for which learning can be broken into two parts: learning the gates, and learning the weights given the gates, the latter of which being described by the Neural Path Kernel. The paper introduces a dual architecture, Deep Linear Gated Networks (DLGN) that parameterizes these two processes separately. The DLGN is argued to aid in interpretability of ReLU networks, with a main conclusion being that the neural network is learned path-by-path instead of layer-by-layer.

The reviewers generally found strength in the motivation and perspective and thought that the DLGN could serve as a useful architecture for aiding interpretability. Some reviewers found the presentation hard to follow, and others were not entirely convinced by the ultimate conclusions. Overall, the reviewers opinions were mixed.

I believe the ICLR community would generally find interest in the DLGN and the interpretations it might afford to deep ReLU networks. However, the number and strength of the conclusions obtained in the current analysis are rather weak. The conclusion that networks learn path-by-path instead of layer-by-layer was emphasized but the implications were not highlighted, and it remains unclear to me and at least some reviewers what the concrete significance of this observation actually is. Another major claim is that the DLGN recovers more than 83.5% of the performance of state-of-the-art DNNs, but a priori it is not obvious what this number means, or if it is even good or bad performance. A more detailed analysis with additional common baselines, ablations, etc., would really help readers understand the significance of the performance gap.

Overall, this is an interesting direction with significant potential, but for the above reasons I cannot recommend the current version for acceptance.